# HiBC: a publicly available collection of bacterial strains isolated from the human gut

Thomas C. A. Hitch [1], Johannes M. Masson[1,15], Charlie Pauvert [1,15], Johanna Bosch [1], Selina Nüchtern[1], Nicole S. Treichel [1], Marko Baloh[1], Soheila Razavi[1], Afrizal Afrizal[1], Ntana Kousetzi[1], Andrea M. Aguirre[1], David Wylensek [1], Amy C. Coates [1], Susan A. V. Jennings[1], Atscharah Panyot[1], Alina Viehof [1], Matthias A. Schmitz [1], Maximilian Stuhrmann[1], Evelyn C. Deis[1], Kevin Bisdorf[1], Maria D. Chiotelli [2], Artur Lissin [3], Isabel Schober [3], Julius Witte [3], Thorsten Cramer [4], Thomas Riedel[3,5], Marie Wende[6], Katrin A. Winter [6], Lena Amend [6], Alessandra Riva [7,8], Stefanie Trinh[9], Laura Mitchell[10], Jonathan Hartman [11], David Berry [7], Jochen Seitz [12], Lukas C. Bossert[11], Marianne Grognot [2], Thorsten Allers [10], Till Strowig [5,6,13], Michael Pester [3,14], Birte Abt[3,5], Lorenz C. Reimer[3], Jörg Overmann [3,5,14] & Thomas Clavel [1] ✉

Numerous bacteria in the human gut microbiome remain unknown and/or have yet to be cultured. While collections of human gut bacteria have been published, few strains are accessible to the scientific community. We have therefore created a publicly available collection of bacterial strains isolated from the human gut. The Human intestinal Bacteria Collection (HiBC) (https://www.hibc.rwth-aachen.de) contains 340 strains representing 198 species within 29 families and 7 phyla, of which 29 previously unknown species are taxonomically described and named. These included two butyrate-producing species of *Faecalibacterium* and new dominant species associated with health and inflammatory bowel disease, *Ruminococcoides intestinale* and *Blautia intestinihominis*, respectively. Plasmids were prolific within the HiBC isolates, with almost half (46%) of strains containing plasmids, with a maximum of six within a strain. This included a broadly occurring plasmid (pBAC) that exists in three diverse forms across *Bacteroidales* species. Megaplasmids were identified within two strains, the pMMCAT megaplasmid is globally present within multiple *Bacteroidales* species. This collection of easily searchable and publicly available gut bacterial isolates will facilitate functional studies of the gut microbiome.

The cultivation of human gut bacteria has accelerated in the last years[1–3], providing valuable information on the presence of novel taxa within gut microbiomes. Yet while these published collections of human gut isolates note the novel taxa within their collections[1,4], rarely is this novelty made known by describing and validly naming the taxa[3].

Curating the taxonomic assignment of such collections is essential, as often outdated names are used, or recently described taxa are ignored, causing greater confusion in the current taxonomic sphere[5–8]. Such curated taxonomy ensures strains are correctly assigned, allowing strain-level diversity to be studied[9]. Variation between strains of the

same species can lead to functional shifts that, for example, alter the association with host dietary habits[10,11]. One-way strains can vary due to the presence of mobile genetic elements, which are known to affect the phenotype of the isolates in which they occur[12]. The study of plasmids within the human gut has been limited until recently[13,14].

Accessibility to collections of strains from the human gut is essential for the mechanistic study of microbe-microbe and microbe-host interactions[15–17]. However, accessibility remains problematic, with few strains being deposited in public culture collections and even fewer of the proposed novel taxa being validated. For most of the strains deposited, genomes are available, but their quality is not always high, and metadata such as the source of isolation, cultivation requirements, and full taxonomy are often lacking. The small number of deposited strains limits future confirmatory or comparative studies, while the lack of curated metadata reduces the value of the genomic information. Only the professional acquisition of strains by public collections can ensure that the high-quality standards required for future work are maintained and that valuable strain-associated information is gathered and made freely accessible to researchers and users worldwide[18]. This is particularly relevant for microbiota research to move beyond associations and towards a more mechanistic understanding, as exemplified by the study of *Akkermansia muciniphila*[19,20].

To provide functional insights into the human gut microbiome based on isolates, we sought to create a publicly accessible collection of bacteria isolated from the human gut. Studying this collection of isolates provided insight into the prevalence and diversity of plasmids within the human gut and the presence of megaplasmids. We also highlight health-associated variation within key genera, including novel species which require further in vivo study.

## Results

### A diverse range of key commensal species isolated from the human gut

The Human intestinal Bacteria Collection (HiBC) consists of 340 strains, representing 198 species, isolated from human faecal samples (Fig. 1). It contains 29 families from across the seven dominant phyla in the human gut: *Bacillota* (n = 173 isolates), *Bacteroidota* (n = 95), *Actinomycetota* (n = 46), *Pseudomonadota* (n = 22), *Desulfobacterota* (n = 2), *Fusobacteriota* (n = 1), and *Verrucomicrobiota* (n = 1). Of the 198 species, 29 are novel taxa that we have described and named for validation according to both the SeqCode[5] and the ICNP[21]. High-quality genomes defined as >90% completeness (99.22 ± 1.10%), <5% contamination (0.50 ± 0.77%), >10× genome coverage (380.08 ± 293.97) were generated for all 340 strains. To improve access to the strains, their genomes, information on their cultivation, and isolation source material, we have created the HiBC web interface; https://www.hibc.rwth-aachen.de. The website provides access to the complete collections of 16S rRNA gene sequences, genomes, plasmids, and metadata.

### Description of dominant novel taxa within the human gut

There has been a renaissance in isolation of strains from the human gut in the last decade[1–4,22–25]. To place the HiBC isolates in the context of prior studies, we analysed previously published work in which phylogenetically diverse isolates were cultured. Across eight large-scale isolation studies and repositories, a total of 12,565 strains from the human gut were reported (Fig. 2a, Supplementary Data 1). Of these, 9707 (76.1%) isolates were claimed to be requestable in the original publications, but only 1539 (12.2%) have been deposited to a culture collection to ensure long-term availability. However, this includes 1063 isolates that have only been deposited to the China General Microbiological Culture Collection Centre (CGMCC), which prevents the accessibility of risk group 2 organisms, many of which are, to researchers outside China. Many strains within the IHU collection (Institut Hospitalier Universitaire Méditerranée Infection) are also

inaccessible under the Nagoya protocol due to concerns surrounding the ethical permission of sampling, which has led to the original paper being retracted[26]. This means only 476 bacterial isolates (3.8%) from the human gut are currently accessible to the community. While access to many of the isolates is limited, most isolates have been genomes sequenced (11,498, 91.5%), with most being high quality (10,893, 86.7%). To understand the diversity captured by each study, we dereplicated the genomes to estimate the number of species cultured by each study (Fig. 2b). A 33-fold reduction in diversity was found in some studies. Despite the value of capturing variability at the strain level, the limited number of samples used for isolation suggests redundancy within these collections. Therefore, while many studies claim to have a large collection of isolates, this is an overestimate of the true diversity captured.

Next, we assessed the ability of all isolated strains to capture the diversity present in the human gut microbiota (Fig. 2c). This was achieved by mapping the isolates genomes against a MAG collection for which corresponding relative abundance values were available[27]. The entire landscape of bacteria isolated from the human gut, both from the literature and this study, captured 69.83 ± 18.62% of an individual's microbiota, while the HiBC alone covered 56.51 ± 20.44%. The 29 novel taxa described within this work enhanced this coverage by 3.75 ± 3.99% on average, and represented >10% of the microbiota from 380 people, which peaked at >40% in two samples. Of the 29 novel taxa, 11 species were identified to have a mean relative abundance >0.15% within positive metagenomic samples (Fig. 2d). The most dominant of these was *Ruminococcoides intestinale* (mean relative abundance = 2.84%; n = 3023), followed by *R. intestinihominis* (1.19%; n = 285). Although only recently described[28], these results suggest that *Ruminococcoides* is a dominant but understudied genus within the human gut. A second genus of clear ecological importance to the human gut is *Lachnospira*, in which we describe two dominant novel species. Interestingly, the two samples in which novel taxa accounted for >40% relative abundance were dominated by either *R. intestinale*, or the novel species *L. rogosae*, identified as a medium priority, HMP most wanted taxa (Supplementary Results). Further study of these species is needed to understand their impact on host health and identify the mechanisms, which is now enabled by access to isolated strains.

To uncover associations of the novel taxa with human health conditions, we studied the ecological occurrence of each protein within *R. intestinale* CLA-AA-H216 (=DSM 117897) (Fig. 2e), as the most abundant novel taxa, and *Blautia intestinihominis* CLA-AA-H95[T] (=DSM 111354) (Fig. 2f), as a member of an important genus associated with both health and disease conditions. The association of individual proteins with disease states was determined using InvestiGUT[29]. This involves studying the prevalence of each individual protein encoded by the strain's genome, or plasmid, across thousands of metagenomes, then statistically determining if they occur significantly more or less frequently in the gut metagenomes of people with a specific condition or healthy controls. Out of the 2167 proteins encoded by *R. intestinale* CLA-AA-H216, comparison between Crohn's disease (CD) patients and healthy controls identified a total of 1902 differentially prevalent proteins, with 1883 being significantly more prevalent in healthy samples, whilst 19 were more prevalent in CD samples. The same pattern was observed with Ulcerative colitis (UC), where a total of 1715 significantly differentially prevalent proteins were identified, with 1684 enriched in healthy samples, and 31 enriched in UC samples. Pathway analysis of the health-associated proteins identified multiple anti-inflammatory pathways, including the ABC transporter for spermidine (PotB/C/D), an anti-inflammatory polyamine[30], and biosynthesis of biotin, which has been shown to be immunomodulatory[31,32], ameliorating colitis[33].

The association of *Blautia* spp. with inflammatory bowel diseases is more complex, as some species within this genus have been shown to ameliorate colitis[34], leading to their inclusion within

 

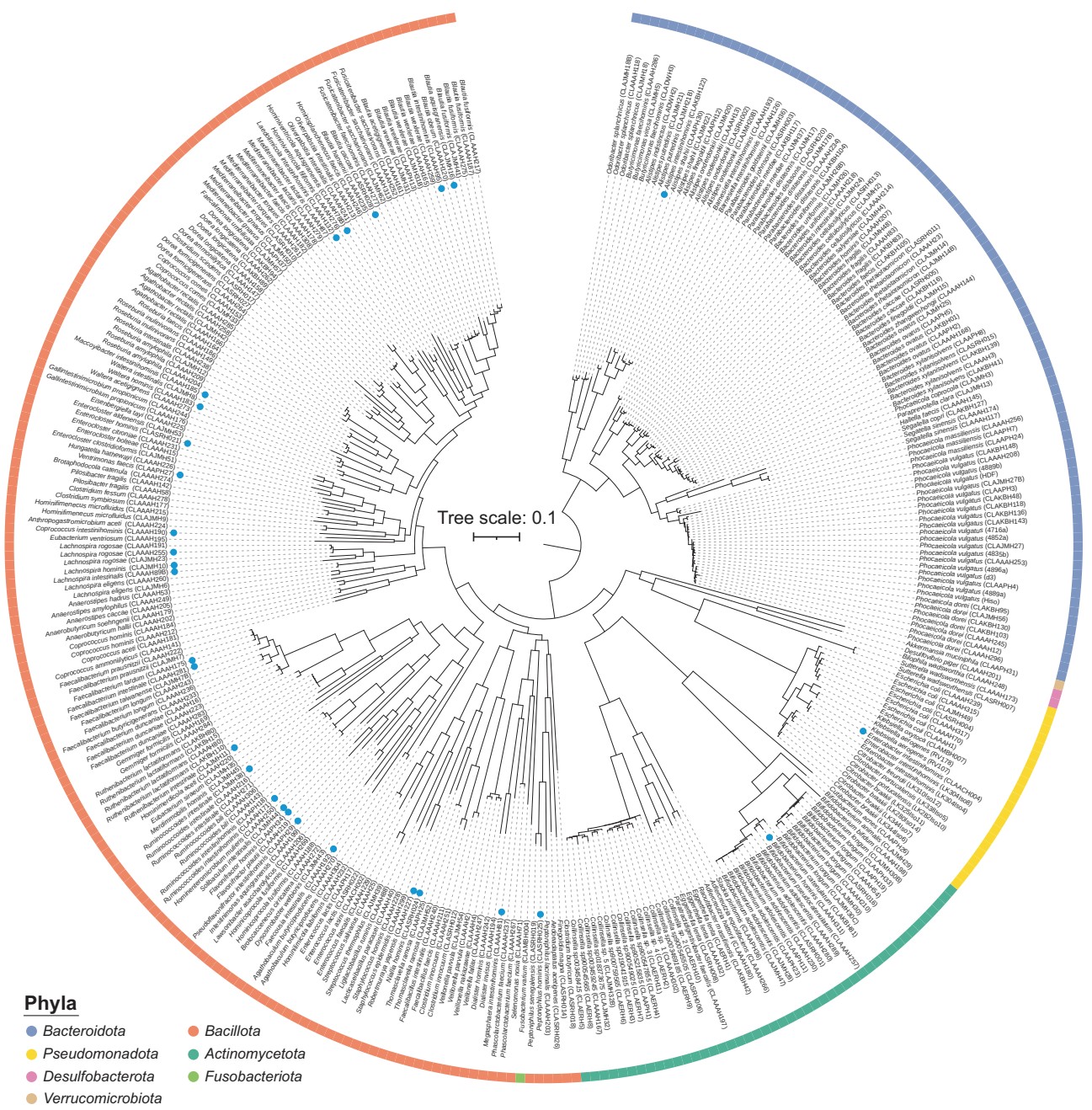

**Phyla**
- *Bacteroidota*
- *Bacillota*
- *Pseudomonadota*
- *Actinomycetota*
- *Desulfobacterota*
- *Fusobacteriota*
- *Verrucomicrobiota*

**Fig. 1 | Phylogenetic diversity of isolates within HiBC.** Tree based on all 340 genomes, generated using Phylophlan[100]. Phyla are indicated with colours. The *Bacillota* are split due to the placement of *Fusobacteriota*, which separated strains assigned to 'Bacillota_A' by GTDB, however, this is dependent on the method used for tree creation (Supplementary Results). The potential need for splitting the phylum *Bacillota* is therefore independently supported by the Phylophlan tree and GTDB. Blue circles identify strains belonging to the 29 novel species that are taxonomically described in this paper.

therapeutic products[35]. However, recent studies have suggested that some *Blautia* spp. may exacerbate colitis[36]. Given the complex interaction of this genus with inflammatory bowel diseases, we studied the association of the novel species, *B. intestinihominis* CLA-AA-H95[T], with both CD and UC. Out of the 3906 proteins, comparison between CD patients and healthy controls identified 523 differentially prevalent proteins, with 488 being significantly more prevalent in healthy samples, whilst 35 were more prevalent in CD samples. In contrast, comparison between UC patients and healthy controls identified 1466 proteins, with 16 being significantly more prevalent in healthy samples, whilst 1450 were more prevalent in UC samples. The functionality of the large number of proteins enriched

within UC samples was studied further, identifying ABC transporters for branched-chain amino acids (LivF/G/H/K/M)[37], phosphate (PstA/B/C/S)[38], and adenosine (NupA/B/C, BmpA)[39], each of which has been linked to colitis.

## Plasmid landscape in the cultured strains and identification of prevalent megaplasmids

Plasmids are present in the human gut and vary between geographically distinct populations[12]. However, most analyses have been culture-independent, preventing accurate taxonomic assignment[13,14,40]. Even studies on horizontal gene transfer using isolates have not reconstructed plasmids and accounted for their impact[2].

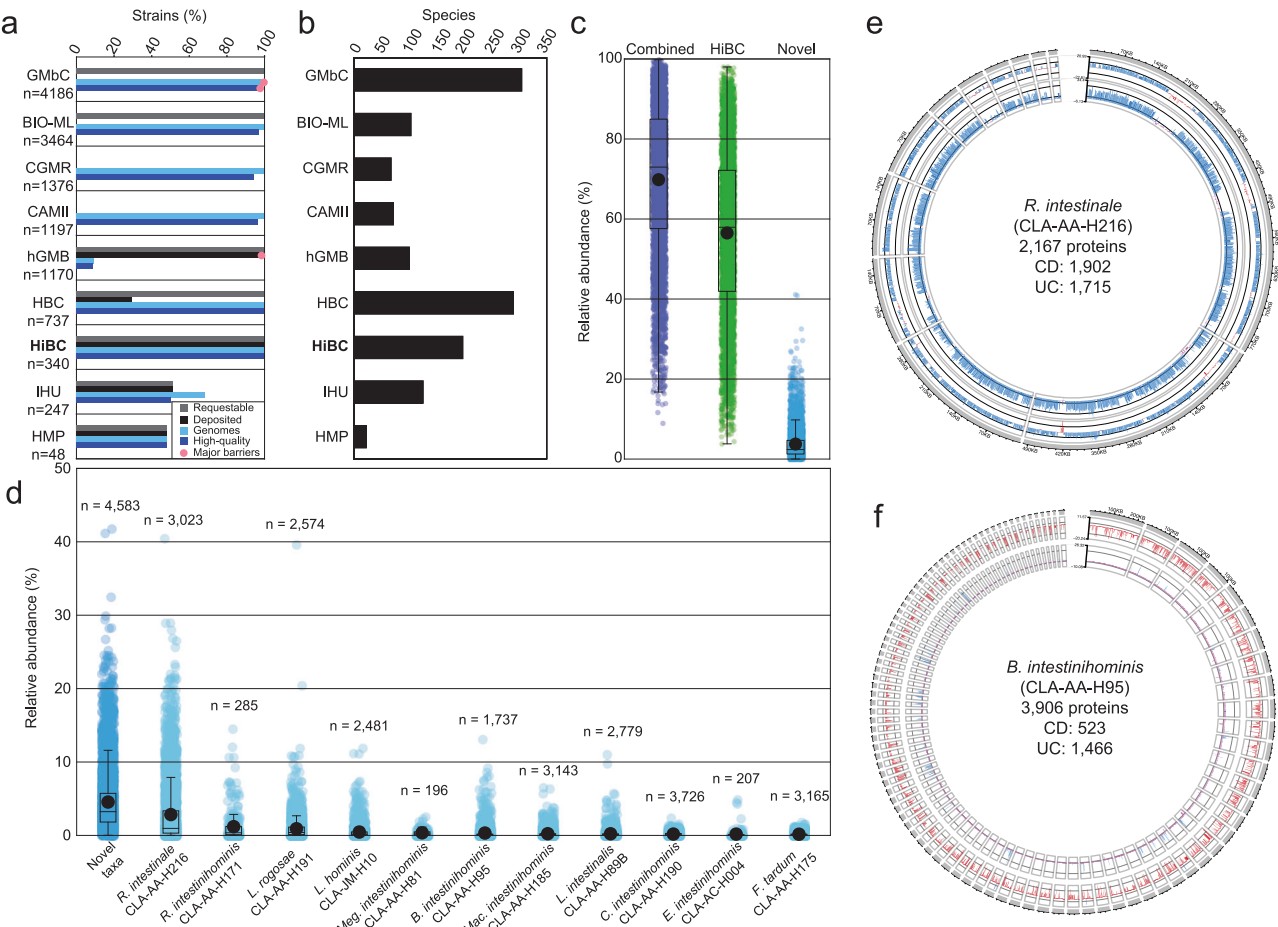

**Fig. 2 | Ecology of isolated human gut bacteria and their proteins. a** The number of strains and genomes produced by eight major isolation projects, along with HiBC, were compared. Strains were deemed requestable if it was claimed in the original publication, although these claims were not substantiated. They were deemed deposited if culture collection identifiers were included in the original paper and were confirmed to exist. Genomes were deemed high quality if they were >90% complete and <5% contaminated. The number of strains within each study is stated, while the percentage meeting each criterion is plotted. Red dots highlight datasets which have barriers to their accessibility, i.e., data available upon request or access limited to specific countries. Strain collections: GMbC, Global Microbiome Conservancy[2]; BIO-ML, Broad Institute-OpenBiome Microbiome Library[24]; CGMR, Chinese Gut Microbial Reference[25]; CAMII, Culturomics by Automated Microbiome Imaging and Isolation[26]; hGMB, Human Gut Microbial BioBank[3]; HBC, Human Gastrointestinal Bacterial Collection[1]; HiBC, Human Intestinal Bacteria Collection (this study); IHU, collection of the Institut Hospitalier Universitaire Méditerranée Infection[27,103]; HMP, Human Microbiome Project at ATCC. **b** Number of species per isolate collection, either via manual curation (HiBC) or dereplication of the available genomes (ANI values > 95% indicated identical species). **c** The

cumulative relative abundance of gut metagenomes across 4624 individuals from Leviatan et al.[28] covered by all isolated bacteria across studies including HiBC (Global isolates, dark blue), HiBC alone (green), or the subset represented by the 29 novel taxa described in this work (light blue), which had matches within 4583 of the samples. **d** Relative abundance of dominant (mean relative abundance >0.25%) novel taxa across 4,624 individuals, with the number of positive samples stated. Each strain represents a distinct novel species, described in detail in the protologues at the end of the "Methods" section. **e, f** Genomic location of proteins significantly differentially prevalent between Crohn's disease (CD) samples and healthy controls (inner ring), or ulcerative colitis (UC) samples and healthy controls (outer ring). The delta-prevalence (prevalence in healthy donors – prevalence in corresponding patients) is shown in blue (more prevalent in healthy controls), red (UC), or mauve (CD). The species, strain, number of proteins predicted within the genome, and those significantly differentially between health conditions are shown within the circle. Only contigs >10 kp were plotted. In panel c and d, boxplots include a line in the centre indicating the median, the boxes represent the interquartile range, and the whiskers represent the minimum and maximum values, not including outliers.

Analysis of bacterial isolates has uncovered many novel plasmids from non-human primates[41]. As many bacterial genome assembly workflows do not consider plasmids, we developed a genome pipeline that first searches for the presence of plasmids using Recycler[42] and plasmidSPAdes[43], assembles them, and then removes their reads from consideration during genome assembly (see "Methods" section). This resulted in the reconstruction of plasmids from 46% of the strains (155 out of 340), with a total of 266 plasmids (Fig. 3a). Plasmids were identified in all phyla except *Fusobacteriota*, for which the HiBC includes only a single isolate. Almost half of the plasmid-positive strains contained more than one plasmid (64 out of 155 strains), up to 6 plasmids in a strain of *Phocaeicola vulgatus* and two strains of the novel species *Enterobacter intestinihominis* (Fig. 3b). Across the 266

plasmids identified in HiBC, 3697 proteins were predicted, although only 639 (17.28%) could be functionally annotated. Of these, nine were antibiotic resistance genes, four of which were copies of the *APH(2″)-IIIa* aminoglycoside resistance gene found on plasmids of identical size (7686 bp) in four strains of *Ruthenibacterium lactiformans* (strains CLA-KB-H110, CLA-KB-H15, CLA-KB-H80, CLA-AA-H80).

Megaplasmids (>100 kp) have been described to occur in *Lactobacillaceae*[44] and *Bifidobacterium breve*[45], but their occurrence in a broader range of commensals from the human gut is unknown. We evaluated the length of the reconstructed plasmids and found the average length was 11.74 ± 17.25 kb (Fig. 3c). This included the identification of two megaplasmids, one within a strain of the

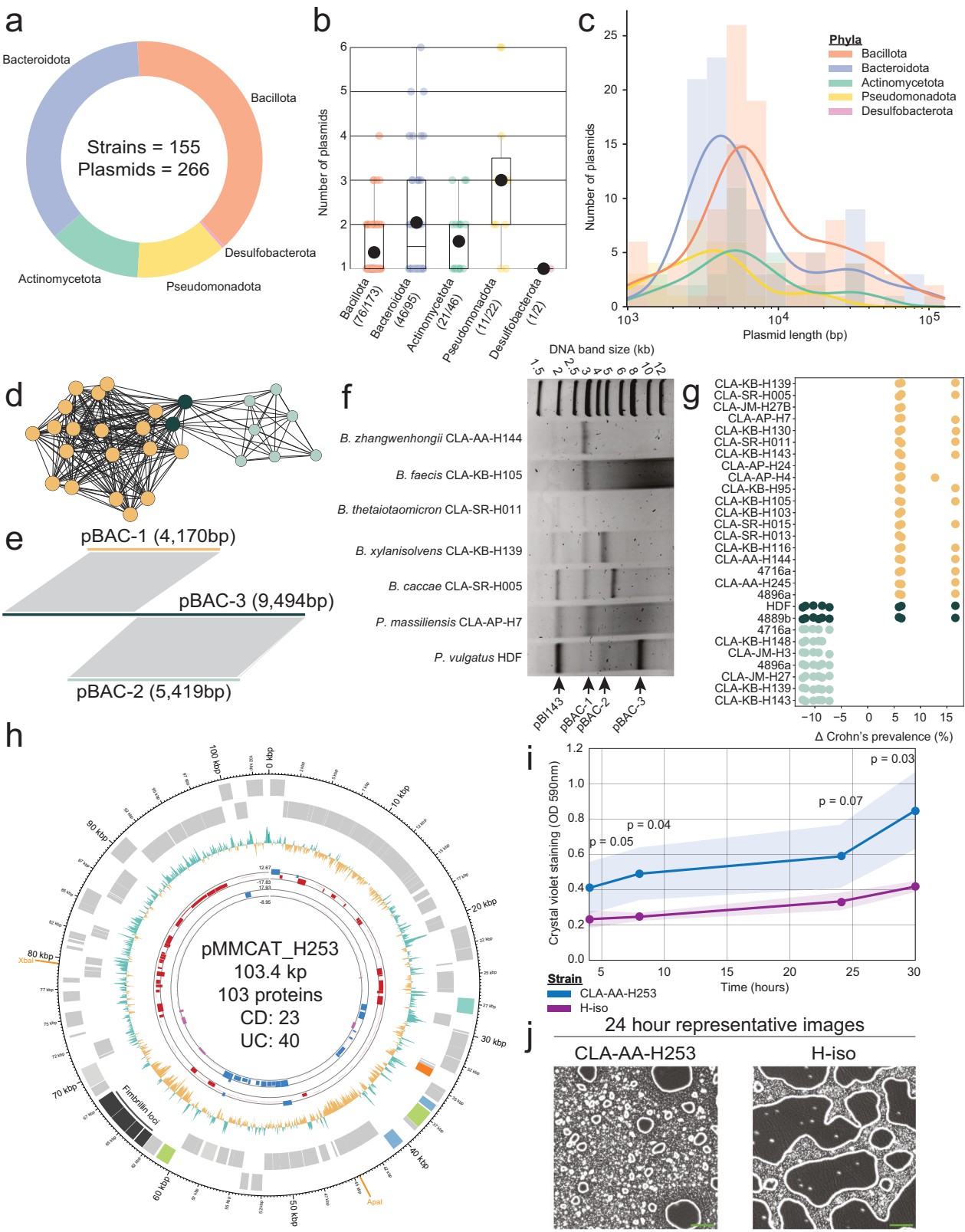

recently described species *Hominifimenecus microfluidus* (strain CLA-JM-H9, =DSM 114605; 125.6 kb) and the other in a strain of *Phocaeicola vulgatus* (strain CLA-AA-H253, =DSM 118718; 103.4 kb). Given the number and size of plasmids a single strain can contain, we assessed their impact on species assignments based on average nucleotide identity, but confirmed that plasmids had only a minor

effect on the taxonomic assignment of a genome (Supplementary Results).

Sequence comparison with MobMess[14] identified 168 plasmid clusters, the two largest of which were specific to the *Bacteroidales* (Supplementary Fig. 1a). The largest cluster shared similarities with 1266 plasmids within the comprehensive plasmid database, PLSDB[46], but

**Fig. 3 | Plasmid repertoire of human gut bacterial isolates and their association with disease and phenotypes. a** Phylum level diversity of plasmid-positive isolates. **b** Number of plasmids reconstructed within each isolate. The boxplots include a central line indicating the median, boxes represent the interquartile range, and whiskers represent the minimum and maximum values, not including outliers. **c** Length of reconstructed plasmids in log-10 scale; a bar represents the number; a line represents the distribution. **d** Network analysis of pBAC plasmid sequence similarity to each other as determined by MobMess. **e** Sequence alignment of a representative from pBAC cluster 1, 2 and 3. Grey lines show regions of >99% similarity. **f** The plasmid repertoire purified from 7 strains predicted to contain either a single or multiple pBAC plasmids, as well as additional plasmids. The arrows indicate bands with a size matching the reconstructed pBAC plasmids. **g** Proteins encoded on pBAC plasmids from different strains with significantly different prevalence between Crohn's disease (CD) patients and healthy controls. Each point represents a single differentially prevalent protein, coloured based on the pBAC cluster the protein was encoded on (**e**). The strains are: *Bacteroides xylanisolvens* (CLA-KB-H139, CLA-SR-H015); *Bacteroides caccae* (CLA-SR-H005, CLA-KB-H116); *Phocaeicola vulgatus* (CLA-JM-H27B, CLA-JM-H27,CLA-KB-H143, CLA-AP-H4,4716a, 4896a, HDF, 4889b, CLA-KB-H148); *Phocaeicola massiliensis* (CLA-AP-H7, CLA-AP-H24); *Phocaeicola dorei* (CLA-KB-H130,

CLA-KB-H95, CLA-KB-H103,CLA-AA-H245); *Phocaeicola coprocola* (CLA-JM-H3); *Bacteroides thetaiotaomicron* (CLA-SR-H011); *Bacteroides faecis* (CLA-KB-H105); *Bacteroides cellulosilyticus* (CLA-SR-H013); *Bacteroides zhangwenhongii* (CLA-AA-H144). **h** Plasmid map of pMMCAT_H253. The innermost rings represent the association of proteins differentially prevalent between CD samples and healthy controls (first ring), or ulcerative colitis (UC) samples and healthy controls (second ring) via InvestiGUT[30]. Proteins enriched in CD are purple, those enriched in healthy samples are blue, and those enriched in UC are red. The third ring represents the GC content relative to the average of the entire plasmid. Boxes are used to represent genes identified on the plasmid in the forward (outer ring) and reverse (inner ring) strand, with coloured boxes being assigned a COG category, while grey boxes represent COG-unassigned proteins. The fimbriae loci proteins are indicated in dark grey. Enzymes with a single restriction site are indicated on the outer ring in orange. **i** Quantification of cell adhesion from a pMMCAT-containing strain of *P. vulgatus* (CLA-AA-H253) and its closest relative strain (H-iso) without the megaplasmid (see "Methods" section). Visualised are the mean values with the 95% confidence intervals. Statistics: paired, one-tailed *t*-tests. **j** Representative images of cell adhesion of the two selected strains. Images were taken from the three replicates tested. Green scale bars represent 50 μm.

lacked an assigned name, hence, we named it pBAC (plasmid *Bacteroidales*). The second largest plasmid cluster was identified as the cryptic plasmid, pBI143, which was recently described to be associated with IBD[13] and occurred in 14 isolates across eight species. Given that pBAC matches 1266 plasmids within PLSDB, while pBI143 matched only 719, it may be the most prolific plasmid in the human gut. We therefore investigated pBAC further, observing it has been most frequently found in the USA but has also been detected in Denmark, Japan, Ireland, and China (Supplementary Fig. 1b). Investigation of the sequence similarity network identified that pBAC occurred in both a shorter (pBAC-1; ~4220 bp) and longer (pBAC-2; ~5419 bp) form, with the two cluster being connected by two 9494 bp variants (pBAC-3) (Fig. 3d). Comparison of representative sequences from each of these three clusters uncovered that both, pBAC-1 and pBAC-2 shared no similarity, but were identical to regions of pBAC-3 (Fig. 3e). These results suggest that the different forms of pBAC share a common origin, implying they can co-occur within the same strain. We identified *B. xylanisolvens* CLA-KB-H139 as a strain which was predicted to contain both pBAC-1 and pBAC-2 and confirmed their existence within the isolate by extracting plasmids from a diverse range of species predicted to contain different forms of pBAC (Fig. 3f). This analysis confirmed that pBAC occurs across the *Bacteroidales*. Given the prevalence of pBAC within common denizens of the human gut, we aimed to study their functional potential. All three pBAC clusters encode mobilisation machinery (MbpA/B/C)[47]. Interestingly, both pBAC-1 and pBAC-2 encode different toxin-antitoxin pairs, with that in pBAC-1 resembling YoeB and YefM[48], while the pBAC-2 encoded system shared similarity with RelE and Phd[49]. Given their differing protein content, we hypothesised their association with host health may differ, hence studied the prevalence of their encoded proteins across disease cohorts (Fig. 3g). pBAC-1 was observed to encode 2–3 proteins that were significantly more prevalent within CD patients, with the most enriched protein being a replication protein, followed by YefM-like and YoeB-like proteins. Conversely, pBAC-2 plasmids encoded 5–6 proteins within significantly lower prevalence in CD patients, with the most being the RelE-like and PhD-like proteins. Toxin-antitoxin systems within *Bacteroidota* plasmids have previously been reported, although pBAC shared no similarity with these described plasmids[50]. These results suggest that the pBAC-encoded toxin-antitoxin pairs are associated with human health, potentially by altering the fitness of their host strain.

In addition to the highly prevalent pBAC plasmid, we studied the megaplasmid present in *P. vulgatus* (CLA-AA-H253), a prevalent and dominant species in the human gut. The megaplasmid matched 26 plasmids within PLSDB, including multiple designated as pMMCAT (Supplementary Fig. 1a), a recently proposed large plasmid from *Bacteroidales*, which has yet to be confirmed using sequencing-independent methods[51], but has been shown to impact its host's ability to form

biofilms[52]. As such, the *P. vulgatus* megaplasmid was designated pMMCAT_H253. PacBio sequencing generated a complete genome for strain CLA-AA-H253, which confirmed the reconstruction of pMMCAT_H253. To provide sequencing-independent validation of pMMCAT_H253, we used pulsed field gel electrophoresis which showed a band of ~100 kb after treatment using XbaI, for which the plasmid had a single restriction site allowing for its linearisation (Supplementary Fig. 2a). The bands' identity was confirmed using southern blot with primers designed based on the predicted pMMCAT_H253 sequence (Supplementary Fig. 2b). Out of the 104 proteins encoded on pMMCAT_H253, only 6 could be assigned to a functional category, highlighting the need for further characterisation of these proteins (Fig. 3h). A locus containing four proteins assigned to 'fimbrillin family proteins' was identified within a region of lower GC content compared to the plasmids average, which may suggest this region represents variable cargo and not the backbone. The presence of 'fimbrillin family proteins' on pMMCAT_H253 was further investigated as fimbriae can facilitate adhesion of cells[53], including to the host epithelium, to enhance colonisation[54]. We therefore studied the ability of CLA-AA-H253, the pMMCAT-containing strain, and its closest related strain (H-iso, 98.9% ANI), which lacks pMMCAT, to adhere to plastic wells. CLA-AA-H253 adhered to the plastic significantly more than H-iso after only 4 h, with 2-fold more bacteria adherent after 30 h of growth (Fig. 3i). The adherence of strain CLA-AA-H253 was observed in the wells, while H-iso was adhered in patches across the wells (Fig. 3j). These results support the need for greater study of pMMCAT as a potential source for phenotype variation within genera known to contribute to host health[55,56].

## Enhanced diversity of *Faecalibacterium* species

The genus *Faecalibacterium* currently (January 2025) contains seven validly published and correctly named species. HiBC contains representatives for five of these, and isolates representing two novel species. Initial 16S rRNA gene analysis assigned these novel species as *Faecalibacterium prausnitzii*, a common commensal of the gut microbiota that has been associated with human health conditions ranging from reducing inflammation[57] to improving cognitive function in Alzheimer's disease[58]. Many of these beneficial observations have been attributed to *F. prausnitzii*'s ability to produce large amounts of the short-chain fatty acid butyrate[59].

Taxonomically, the genomic diversity of *F. prausnitzii* is proposed to be greater than suggested by 16S rRNA gene sequence analysis[60]. This has led the GTDB to split this species into 12 proposed species. HiBC contains representatives of the not yet described species defined by GTDB-Tk as *F. prausnitzii* A (strain CLA-AA-H175) and *F. prausnitzii* J (strain CLA-AA-H281). This was confirmed using ANI and can be observed in a phylogenomic tree (Fig. 4a). We therefore propose to

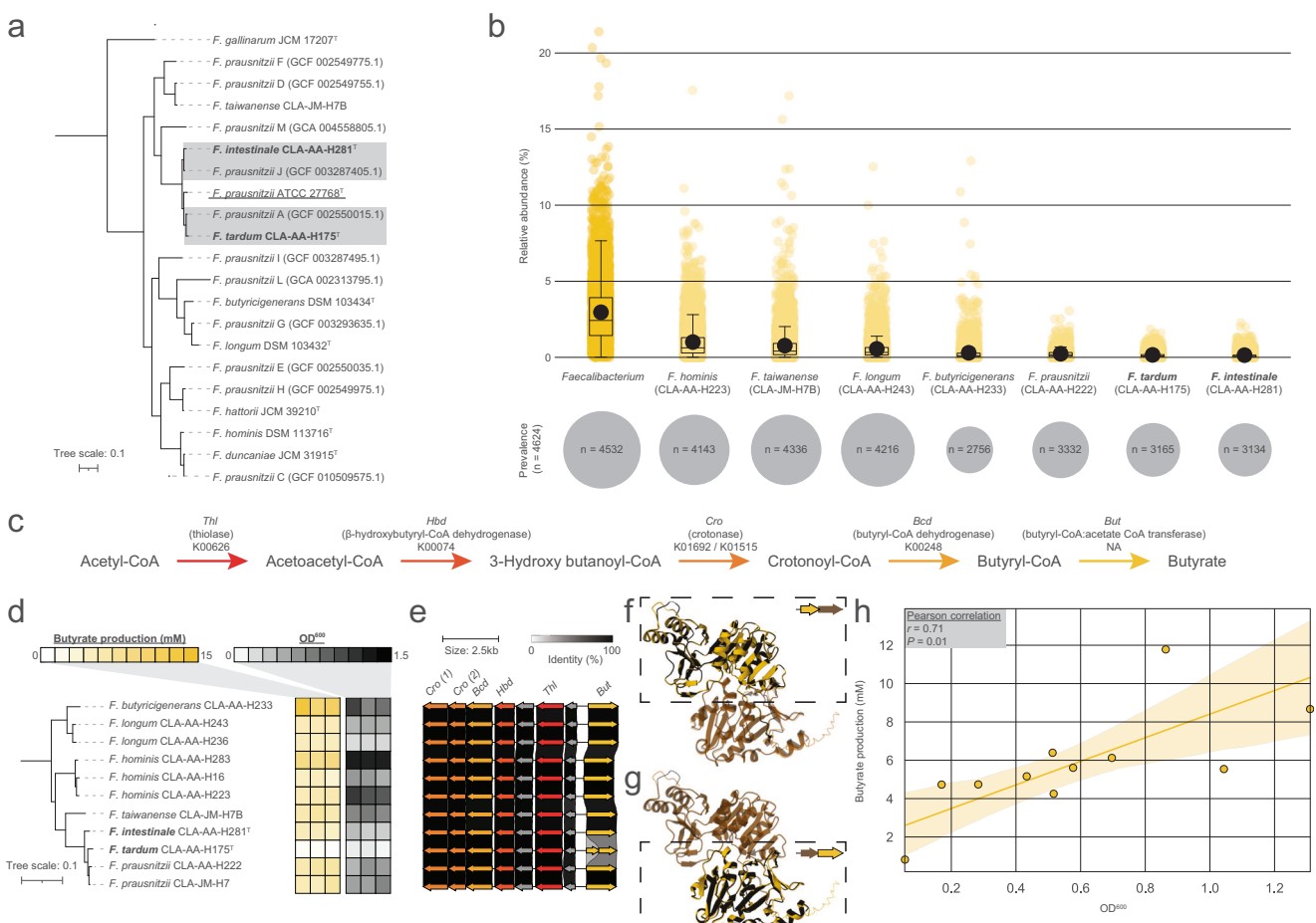

**Fig. 4 | Novel diversity within *Faecalibacterium* and strain-dependent butyrate production. a** The two novel species of *Faecalibacterium* described within this paper placed within the current landscape of *Faecalibacterium* spp. with a valid name, along with the type genomes for proposed divisions of *F. prausnitzii*, as determined by GTDB. The phylogenomic tree was rooted on *Ruminococcus bromii* ATCC 27255[T]. Novel species are in bold, and the type strain of *F. prausnitzii* is underlined. **b** Relative abundance and prevalence of the genus *Faecalibacterium*, and each *Faecalibacterium* species represented within HiBC across 4624 metagenomic samples. The boxplots include a central line indicating the median, boxes represent the interquartile range, and whiskers represent the minimum and maximum values, not including outliers. **c** Butyrate production pathway in *Faecalibacterium* with gene names and KEGG ortholog identifiers when possible. **d** Phylogenomic tree of the *Faecalibacterium* strains within HiBC, displaying the ability of each strain to produce butyrate over a 48 h-period, along with the OD600

that the strain achieved during the testing period (*n* = 3 independent batch cultures for each strain; the replicates are shown with individual boxes). The phylogenomic tree was rooted on *R. bromii* ATCC 27255[T]. **e** Sequence comparison of the butyrate production loci across the *Faecalibacterium* strains. Genes are coloured based on their assignment to each step in the butyrate production pathway in (**c**). **f** AlphaFold3 model of the *But* complex in *F. tardum* CLA-AA-H175 against the full *But* protein in *F. prausnitzii* CLA-AA-H222. The first CLA-AA-H175 *But* gene is highlighted in yellow in the dashed box, while the second gene is shown in brown. The highlighted protein is indicated in the top right of the dashed box. **g** Same as in (**f**), but this time the second CLA-AA-H175 *But* gene is highlighted in yellow in the dashed box, while the first gene is in brown. The highlighted protein is indicated in the top right of the dashed box. **h** Correlation of the mean OD600 against the mean butyrate production of each strain with a linear regression and its 95% confidence interval and analysed using a two-sided Pearson correlation coefficient.

name these novel species *Faecalibacterium tardum* (CLA-AA-H175) and *Faecalibacterium intestinale* (CLA-AA-H281). Protologues for these, and the other 27 novel taxa described in this paper, are provided at the end of the "Methods" section. Both novel species were observed in the majority of microbiota studied at similar relative abundances as *F. prausnitzii* (*F. prausnitzii* = 0.24 ± 0.24%; *F. tardum* = 0.16 ± 0.20%; *F. intestinale* = 0.14 ± 0.21%) (Fig. 4b).

Butyrate production by *Faecalibacterium* spp. is critical for host health, hence, we studied variability between the strains in their ability to produce butyrate (Fig. 4; Supplementary Data 2). Butyrate production by *Faecalibacterium* is achieved via the acetyl-CoA pathway, which is also the dominant pathway for butyrate production in the human gut (Fig. 4c)[61,62]. While all *Faecalibacterium* strains within HiBC produced butyrate, the amounts varied greatly, with *F. tardum* (strain CLA-AA-H175) producing only 0.8 ± 0.16 mM, whereas strains of *F. longum* and *F. butyricigenerans* produced >10 mM (Fig. 4d). Given that butyrate production is a conserved phenotype of this genus, we

investigated if variation in the acetyl-CoA pathway loci was responsible for the observed variation between isolates (Fig. 4e). While little variation was observed between most of the strains, *F. tardum* CLA-AA-H175 encoded two truncated copies of the butryl-CoA:acetate-CoA transferase gene. To understand the cause of these truncated genes, we studied their placement in relation to each other and identified that they occurred on the same strand but in different frames, with an overlap of 55 bp. Interestingly, the second truncated gene is encoded with 'GTG' as an alternative start codon, which may lead to lower transcription and hence alter butyrate production further[63]. Structural modelling of these proteins using AlphaFold3 identified that the truncated proteins form a complex that shares 100% (Fig. 4f) and 90% (Fig. 4g) identity with the full protein from *F. prausnitzii* CLA-AA-H222, respectively. Acetate utilisation and butyrate production have been linked to the growth of *Faecalibacterium* spp.[64,65], hence we considered whether the ability of the isolate to grow under the testing conditions alters its metabolism, and eventually the ability to

produce butyrate. We therefore correlated the average butyrate production of each strain against its average growth, measured by OD600 (Fig. 4h). A strong, significant correlation ($r = 0.76$; $p = 0.01$) was identified between the growth of a strain and the amount of butyrate it produced. These results suggest that strains of *Faecalibacterium* vary greatly in their ability to produce butyrate, either as a result of genetic modification, as in the case of *F. tardum*, or due to the strain's ability to grow in vitro. This further implies that strains that grow at high abundance in vivo are likely to produce the most butyrate.

## Discussion

Over the last decade, the renewed interest in cultivation has expanded the number of genomes available for taxa within the human gut. However, the corresponding strains are rarely accessible to the research community, as highlighted by only 3.8% of human gut isolates being publicly deposited within a culture collection. The process of strain deposition within culture collections has previously been highlighted as a major limiting factor in making strains publicly available[66]. To overcome this, we initiated a system for bulk deposition of strains at the DSMZ (Supplementary Methods). Initial registration of strains with StrainInfo prior to deposition allows the processing of strains to be monitored. The assignment of DOIs to each strain ensures their traceability, facilitating publication while ensuring strains are consistently referenced. Application of this system allowed for deposition of 340 human gut isolates in the DSMZ, which in turn allowed the naming and description of 29 novel species, including the type species of three novel genera. Based on this, the HiBC is the first collection of human gut isolates that is entirely publicly accessible.

Large-scale cultivation of human gut bacteria in this study led to the isolation of many novel species and genera. Whilst previous cultivation studies report many isolates representing novel diversity, these bacteria are rarely taxonomically described and named. The discovery of prevalent and important novel species in this work will facilitate future experiments on the role of bacteria in health and disease. This included *Ruminococcoides intestinale*, *Blautia intestinihominis*, and the isolation of multiple strains of *Faecalibacterium*, including species with the potential to reduce gastrointestinal inflammation[57]. We confirmed that *F. prausnitzii* represents a diverse group of related species that show reduced 16S rRNA gene sequence diversity[60]. By naming two of these species and describing the butyrate production and growth characteristics of seven different species in this genus, we observed that their ability to produce butyrate was proportional to their ability to grow. Given these results, further quantification of these strains' ability to grow within the human gut is required to understand their contribution to SCFA levels in vivo, and therefore their potential impact on host health.

The reconstruction of plasmids for almost half of HiBC isolates allows the direct link of plasmids to strains of various species, information previously lacking from studies on plasmids from the human gut[12–14]. Of note, *Bacteroidales* strains often contain multiple plasmids, particularly copies of pBI143 or pBAC. The pBAC plasmid was the most frequently reconstructed plasmid and was observed to occur in three forms. Investigation of these forms uncovered that both major forms have differing associations with host health. The megaplasmid pMMCAT was also identified within a strain of *P. vulgatus*, leading to enhanced adhesion to surfaces compared to strains lacking pMMCAT. By making the reconstructed plasmids and strains publicly available, we believe this resource expands the toolbox of methods for genetically modifying non-model commensal gut microbes.

Metagenomic studies have provided important insights into the taxonomic and functional diversity present in the human gut, however, they are unable to provide mechanistic insights. By enabling broad access to human gut isolates in this work, the functionality of taxa can be experimentally validated, facilitating mechanistic studies of microbe-host interactions.

## Methods

### Ethics

The Ethics Committee of the Medical Faculty of RWTH University Aachen permitted bacterial isolation from human stool under ethical numbers EK 23-055, EK 316-16, and EK 194/19. For strains originating from Vienna, isolation was approved by the University of Vienna ethics committee under ethical number 00161. For strains originating from Braunschweig, isolation was approved by the Ethics Committee of Lower Saxony (MHH permit No. 6794, 8629 and 8750). Written informed consent was signed by all enrolled participants.

### Bacterial isolation and cultivation

Human stool was collected in sterile plastic bags and stored in tightly sealed plastic buckets until further processing. An oxygen scavenger sachet (BD Biosciences; ref. 260683) was added to each bucket before sealing to reduce exposure to oxygen. All samples were processed in the lab within 24 h of collection. First, the faecal material was homogenised by manual kneading of the plastic bag. Stool samples were either mixed 1:5 with anaerobic FMT media[67], distributed to 1 mL aliquots and stored at −80 °C until further use, or the samples were further processed as described previously[68]. Hence, approximately 5 g of the samples were dissolved in 50 ml of anoxic PBS supplemented with peptone (0.05% w/v), L-cysteine (0.05% w/v) and dithiothreitol (DTT) (0.02% w/v) by shaking the glass flask intensely. A syringe with needle was used to transfer 5 ml of the slurry through a rubber stopper (previously flamed using ethanol) into a second glass flask containing 45 ml of the same buffer in an anaerobic atmosphere (89.3% $N_2$, 6% $CO_2$, 4.7% $H_2$) to create a 1:100 dilution of the original sample. The flask was moved into an anaerobic workstation (MBraun GmbH, Germany) to prepare 2 ml aliquots under anaerobic conditions, which were mixed with 2 ml of 40% anaerobic glycerol to a final concentration of 20%, then stored at −80 °C until use.

Two different approaches were used for bacterial isolation. For classical isolation, the faecal aliquots were thawed inside the anaerobic workstation and diluted with anoxic PBS (see above) in a tenfold dilution series ($10^{-2}$–$10^{-6}$). 50 μl of each dilution step was transferred onto different agar plates and spread using an L-spatula. Single colonies were picked after 1–7 days of incubation at 37 °C under anaerobic conditions. Bacterial cells were re-streaked at least three times to guarantee the purity of the culture. For the second isolation approach, the single-cell dispenser b.sight (Cytena GmbH, Germany) was used as described before[69]. Details on culture media preparation and ingredients are provided in the Supplementary Methods.

The isolated bacteria were first identified using MALDI-TOF MS (Bruker Daltonics, Bremen, Germany). If a species was not yet present within HiBC, or could not be reliably identified by MALDI-TOF MS, the bacteria were stored as cryo-stocks and DNA extracted for genome sequencing. Isolates from different donors that belonged to the same species were considered to represent individual strains and were kept in the collection. Cryo-stocks were generated by freezing in glycerol media (end concentration 20%) at −80 °C. All HiBC strains have been deposited at the Leibniz Institute-German Collection of Microorganisms and Cell Cultures (DSMZ). In addition, the strains representing novel taxa were deposited at either the Belgian Coordinated Collections of Microorganisms (BCCM) or the Japan Collection of Microorganisms (JCM).

Deposition at the DSMZ was facilitated by developing a bulk-deposition system. In brief, after the DSMZ has assessed a researcher's request to submit a strain collection, it requests the strains in large batches, which are deposited in an iterative process. Once cultures have been received by the DSMZ and passed initial checks, the system allows for the allocation of stable identifiers in the form of Digital Object Identifiers (DOIs) provided by the database StrainInfo (www.strainfo.dsmz.de), where each strain is permanently registered with its metadata. This step ensures that all data (e.g., publications and sequence data) can be linked to the corresponding strain entry when

the strains become available. Strain entries are initially displayed with the status 'Deposition in progress'. When the deposition process is completed (which may take several months) and the strains become available in the DSMZ catalogue, the status of the strains is updated to 'Published' in StrainInfo. This process is detailed in greater depth in the Supplementary Methods.

## Metabolite production analysis

Concentrations of short-chain fatty acids (SCFAs) (acetate, butyrate, propionate, valerate), branched-chain fatty acids (isobutyrate, iso-valerate), intermediate metabolites (ethanol, formate, lactate, 1,2-propandiol, 1-propanol, succinate), as well as mono- and disaccharides (galactose, glucose, lactose) were measured by high-performance liquid chromatography (HPLC). The bacterial strains were grown anaerobically, apart from *Robertmurraya yapensis* CLA-AA-H227, which was grown under aerobic conditions, in YCFA broth (DSMZ Medium No. 1611) in Hungate tubes for 48 h at 37 °C. Triplicate cultures from each strain were measured. Baseline controls were taken from each sterile Hungate tube before inoculation. Each taken sample was centrifuged ($10,000 \times g$, 10 min, 4 °C), supernatants were collected and stored at −80 °C until HPLC measurement. Samples were prepared and measured (including HPLC settings) as described previously[68]. External standards were used for concentration determination by comparison of peak retention times (HPLC grade compounds were purchased from Sigma-Aldrich). Peaks were integrated using the Chromaster System Manager Software (Version 2.0, Hitachi High-Tech Science Corporation 2013, 2017). Metabolite concentrations >0.2 mM (limit of detection (LOD) for citrate, 1-propanol), >0.24 mM (LOD for butyrate, formate, galactose, glucose, isobutyrate, isovalerate, lactose, valerate), >0.4 mM (ethanol, 1,2-propandiol) and >0.8 mM (acetate, lactate, propionate, succinate) were considered for statistical analysis if present in all three replicates. Production and consumption of metabolites were calculated by subtracting the baseline values from the sample taken after 48 h of growth.

## Cellular fatty acids (CFAs) determination

Cellular fatty acids were measured at the Leibniz Institute DSMZ. The strains representing novel taxa were grown under the conditions indicated in the respective protologues. Approximately 100 mg (wet weight) of cell biomass was extracted according to the standard protocol of the Microbial Identification System (MIDI Inc., version 6.1; technical note #101). CFAs were analysed by converting them into fatty acid methyl esters (FAMEs) through saponification, methylation, and extraction. The resulting FAME mixtures were separated using gas chromatography (GC) and detected with a flame ionisation detector (FID). Subsequent analysis involved the identification of fatty acids using a GC-MS system (Agilent GC-MS 7000D) as described by Vieira et al.[70]. Further derivatisation methods were used for structural elucidation of unidentified compounds. For branched-chain fatty acids, cyclo-positions, and multiple double bonds, 4,4-dimethyloxazoline (DMOX) derivatives were analysed[71].

## Isolation of genomic DNA

DNA was isolated using a modified protocol according to Godon et al.[72]. Frozen cell pellets were mixed with 600 µl stool DNA stabiliser (Stratec biomedical), thawed, and transferred into autoclaved 2-ml screw-cap tubes containing 500 mg 0.1-mm-diameter silica/zirconia beads. Next, 250 µL 4 M guanidine thiocyanate in 0.1 M Tris (pH 7.5) and 500 µL 5% N-lauroyl sarcosine in 0.1 M PBS (pH 8.0) were added. Samples were incubated at 70 °C and 700 rpm for 60 min. A FastPrep® instrument (MP Biomedicals) fitted with a $24 \times 2$ ml cooling adaptor filled with dry ice was used for cell disruption. The programme was run 3 times for 40 s at 6.5 M/s. After each run, the cooling adaptor was refilled with dry ice. An amount of 15 mg Polyvinylpyrrolidone (PVPP) was added and samples were vortexed, followed by 3 min centrifugation at $15,000 \times g$ and 4 °C.

Approximately 650 µl of the supernatant were transferred into a new 2 ml tube, which was centrifuged again for 3 min at $15.000 \times g$ and 4 °C. Subsequently, 500 µl of the supernatant was transferred into a new 2 ml tube and 50 µg of RNase was added. After 20 min at 37 °C and 700 rpm, gDNA was isolated using the NucleoSpin® gDNA Clean-up Kit from Macherey-Nagel. Isolation was performed according to the manufacturer's protocol. DNA was eluted from columns twice using 40 µl Elution buffer, and concentration was measured with NanoDrop® (Thermo Scientific). Samples were stored at −20 °C.

## Genome library preparation and Illumina sequencing

Library preparation and sequencing were conducted using the NEBNext Ultra II FS DNA Library Prep Kit for Illumina with dual index primers and ~300 ng of DNA (NEB Inc.) on an automation platform (Biomeki5, Beckman Coulter) according to the manufacturer's instructions. The time used for enzymatic shearing to ca. 500 bp to be used on a MiSeq run was 10 min, and to ca. 200 bp to be used on a NextSeq run was 30 min. PCR enrichment of adaptor-ligated DNA was conducted with five cycles using NEBNext Multiplex Oligos for Illumina (NEB, USA) for paired-end barcoding. AMPure beads (Beckman Coulter, USA) were used for size selection and clean-up of adaptor-ligated DNA. Sequencing was performed either at the IZKF Core Facility Genomics (Uniklinik RWTH Aachen) on a NextSeq platform (Illumina) (PE 150 bp) or on a MiSeq (Illumina) (PE 300 bp) in-house.

## Sequencing data processing

Genomes were assembled from paired-end Illumina short reads. Low-quality reads with an expected average quality below 20 over a 5-base window, containing adaptors, and shorter than 50 bases were discarded using Trimmomatic (v0.39)[73]. Reads with phiX sequences were removed using BBduk from the BBtools suite[74]. Plasmid sequences were reported only when inferred from plasmidSPades (v3.15.5)[75] and then from Recycler (v0.7)[42]. Reads free from plasmid sequences were then assembled using SPades (v3.15.5)[43] with default parameters.

Contigs above 500 bp in assemblies were kept for quality evaluation. Following the MIxS specifications[76] as well as requirements from the SeqCode[5], genomes with a coverage above 10×, a completion above 90%, and a contamination below 5% based on CheckM (v1.2.2)[77] estimation using single-copy marker genes were flagged as high-quality draft genomes. Additional quality flags on the assembly included: (1) <100 contigs, longest contig >100 kp, and N50 > 25 kp using QUAST (v5.0.2)[78]; (2) detectable 16S and 23S rRNA genes using metaxa2 (v2.2.3)[79]; (3) >18 unique essential tRNAs genes and a detectable 5S rRNA gene sequence using the annotations of bakta (v1.6.1)[80].

The genome assembly pipeline, including the reconstruction of plasmids and quality checks steps, is available as a reproducible workflow using Snakemake (v7.9.0)[81]: www.github.com/clavellab/genome-assembly.

## Genome analysis

Taxonomic, functional, and ecological analysis for all genomes was conducted using Protologger (v1.3)[82]. For this, 16S rRNA gene sequences were extracted from each strain's genome using barrnap v0.9 (default settings) (www.github.com/tseemann/barrnap), and the longest 16S rRNA gene sequence from each genome was used as input for Protologger. Phylogenomic trees were generated using Phylophlan v3.0.60[83] with proteomes predicted using Prodigal v2.6.3 (default settings)[84]. Gut metabolic modules presence was based on the identification of all included KOs within a genome, detected by Kofamscan v1.3.0[85].

## Comparison to published isolate collections

Metadata for a previously published collection of microbial isolates from the human gut were obtained from screening literature.

To prevent redundancy, we have excluded articles that are a subset of large collections[4], and those that lack strain metadata or genome access[86]. In the case of isolate collections from multiple body sites, the isolation source was checked to be related to the gastrointestinal tract. Of the 12,565 strains identified, 11,498 had genomes and hence could be analysed. Of these, 10,893 had a high-quality genome and were studied. High quality was determined by completion above 90%, and less than 5% contamination based on CheckM (v1.2.2). Strains were determined to be 'publicly available' if a genome is available and the strain has been deposited within a culture collection, i.e., DSMZ, CGMCC, etc. The validity of each published culture accession number was checked to ensure strain availability. Genomes within the collections were dereplicated at the species level based on >95% ANI values via FastANI[87].

### Ecological analysis

The occurrence of each strain across 16S rRNA gene amplicon samples was conducted using Protologger[82]. In essence, 1000 16S rRNA amplicon samples for each body site (gut, vagina, skin, lung) were processed by IMNGS[88]. Next, their operational taxonomic unit (OTU) sequences were compared against the 16S rRNA gene sequences of the HiBC strains using BLASTN (>97% identity, >80% coverage). Comparison of HiBC genomes against metagenome-assembled genomes (MAGs) was also conducted using Protologger and its collection of 42,927 MAGs with curated metadata. The 16S rRNA gene sequences of the Human Microbiome Projects "most wanted" taxa were obtained from Fodor et al.[89] and their priority taken from the original publication[89]. The most wanted sequences were compared to the 16S rRNA gene sequences of the HiBC strains using BLASTN (>97% identity, >80% coverage)[90]. The prevalence and relative abundance of the HiBC strains within shotgun metagenomic samples were determined by comparison of the HiBC strain genomes to the representative genomes from Leviatan et al.[27] via FastANI (>95% ANI)[87]. HiBC strains matching to representative genomes were then connected to the pre-calculated relative abundance of the representative genomes across 4,623 individuals, published in Leviatan et al.[27].

### Plasmid analysis

Sequence similarity between the isolates plasmids was determined using MobMess[14] and clusters were visualised with Cytoscape[91]. Clustering was conducted first based only on the isolate plasmids, and then including the background previously predicted human gut plasmids. The similarity of each plasmid to known sequences was determined with the 'mash dist' option of PLSDB (v2023_11_03_v2)[46]. The ecology of each plasmid was inferred from the geographical location assigned to each matching plasmid. The sequences of pBAC plasmids were rotated using Rotate, and then aligned with Easyfig[92]. Only regions of similarity with an identity >90% were studied.

### Pulsed field gel electrophoresis

Cultures were grown overnight in 5 mL BHI media, centrifuged at 2400 × $g$ for 5 min (2 mL for *P. vulgatus* CLA-AA-H253, 5 mL for *H. microfluidus* CLA-JM-H9), resuspended in 100 μL sterile PBS and incubated at 45 °C for 10 min. Of this, 80 μL was mixed with 320 μL 1% agarose (Bio-Rad Certified Megabase Agarose) at 45 °C, 80 μL transferred to PFGE plug moulds (Bio-Rad CHEF Disposable Plug Molds, 50-Well) and cooled at 4 °C for 30 min. Solidified agarose plugs were transferred to lysis solution inside 2 mL microcentrifuge tubes (20 mM Tris-HCl pH 8.8, 500 mM EDTA pH 8, 1% N-laurylsarcosine, 1 mg/mL Proteinase K) and incubated with shaking at 250 rpm at 45 °C for 4 h. Lysis solution was replaced with fresh lysis solution + RNAse (10 μg/mL) and incubated with shaking at 45 °C overnight. Lysis solution was replaced with wash solution (25 mM Tris-HCl pH 7.5, 100 mM EDTA pH 8) and incubated with shaking at 45 °C. Plugs were then transferred to fresh 2 mL microcentrifuge tubes with wash solution containing phenylmethanesulfonyl fluoride (1 mM) and incubated with shaking at 45 °C for 30 min, twice. Plugs were then transferred to 1 mL wash solution at stored at 4 °C until use in PFGE. For analysis of intact genomic DNA, agarose plugs were subjected to 100 Gy of γ radiation using a $^{137}$Cs source (Gammacell 1000), to linearise circular chromosomes[93]. PFGE was performed using a CHEF Mapper apparatus (Bio-Rad). Intact and XbaI-digested DNA fragments were separated on a 1.2% agarose gel in 0.5× TBE at 14 °C, with a gradient voltage of 6 V/cm, linear ramping, an included angle of 120°, initial and final switch times of 0.64 s and 1 min 13.22 s, respectively, and a run time of 20 h 46 min.

### Southern blot

The DNA probe was designed by identifying a 1428 bp DNA sequence unique to the pMMCAT_H258 plasmid sequence. Primers sequences (forward primer: GAATTAACGGCCGAGTTCGC; reverse primer: TACCAGTACCGAGGGGAAGG) were checked against the *P. vulgatus* genome sequence using SnapGene (www.snapgene.com) to confirm specificity to only plasmid DNA. PCR primers were then designed based on the regions flanking the unique region. *P. vulgatus* total genomic and plasmid DNA was extracted using the Mericon DNA Bacteria Plus Kit following the manufacturer's instructions (Qiagen). This DNA was used as a template to amplify the target DNA sequence using Thermo Fisher Scientific Phusion Hot Start 2 DNA polymerase and the above primer set. The amplified DNA was visualised by gel electrophoresis to confirm the expected DNA size, and the DNA was extracted and purified using Thermo Fisher Scientific GeneJET gel extraction kit.

The Southern blot gel was stained with 0.5 μg/ml ethidium bromide and visualised, then acid-nicked in 0.25 M HCl, and subsequently denatured in 1.5 M NaCl, 0.5 M NaOH. The DNA was transferred onto a GE Healthcare Amersham Hybond XL membrane by vacuum transfer using a Vacugene XL gel blotter (Pharmacia Biotech) for 1 h at 40 mBar. The membrane was briefly neutralised in 2× SSPE (20× SSPE: 3 M NaCl, 230 mM NaH2PO4, 32 mM EDTA, pH 7.4) and the DNA then crosslinked with 120 mJ/cm2 UV. The membrane was pre-hybridised for 4 h at 65 °C in 6× SSPE, 1% SDS, 5× Denhardt's solution (100× Denhardt's solution: 2% Ficoll 400, 2% polyvinyl pyrrolidone 360, 2% bovine serum albumin), 200 μg/ml salmon sperm DNA (Roche, boiled). The DNA probe used 50 ng of DNA template and was radiolabelled using 0.74 MBq of [α-32P] dCTP (Perkin Elmer) and HiPrime random priming mix (Roche), then purified on a Bio-Rad P-30 column. The membrane was hybridised with the radiolabelled DNA probe overnight at 65 °C in 6× SSPE, 1% SDS, 5% dextran sulphate, 200 μg/ml salmon sperm DNA (Roche, boiled). The membrane was then washed twice for 30 min at 65 °C in 2× SSPE, 0.5% SDS, and twice for 30 min at 65 °C in 0.2× SSPE, 0.5% SDS, before being exposed to a phosphorimager screen (Fujifilm) for 24 h and then scanned on a GE Healthcare Typhoon.

### Plasmid extraction and visualisation

Frozen cryo-stocks were plated on mGAM plates and incubated anaerobically at 37 °C for 48 h. A single colony was inoculated into 5 mL of BHI media within a Hungate tube and incubated for 48 h. 2 mL of culture was transferred into 2 mL microcentrifuge tubes and centrifuged at 18,000 × $g$ for 5 min. The medium was removed and the pellet processed using QIAprep Spin Miniprep kit, following the manufacturer's recommendations (Qiagen, Cat. No. 27106). Depending on DNA concentration, 10–20 μL of extracted DNA was run on a 0.5% agarose gel (Sigma-Aldrich A9539) at 80 V for 120 min (Bio-Rad Biometra) along with GeneRuler 1 kb DNA ladder (Thermo Fisher Scientific SM0312) and stained with a dye (Midori Green Advance, BulldogBio). The gel was imaged with the Bio-Rad GelDoc Go imaging system.

## Crystal violet staining to quantify strain adhesion

Strains of interest were tested in triplicate. For each, one colony was picked and grown to saturation overnight in BHI. Cultures were diluted back to an OD600nm of 0.1 in fresh BHI, 3 ml were placed per well in 6-well plates (Nunc cell-culture, treated, flat clear polystyrene plates, Thermo Scientific), and the plates were incubated in static conditions at 37 °C. For each time point, a plate containing the two strains to be compared was removed from the anaerobic cultivation chamber, rinsed with water, and dried at 60 °C for 2 h. Each well was incubated with 1% (w/v) crystal violet solution, then rinsed 3 times with deionised water and air dried. Once all plates were collected, wells were destained by adding 1 ml of a 1:4 acetone:ethanol solution. After gentle mixing, at least three technical replicates of 200 μL per well were placed in wells of a 96-well plate. The optical density of crystal violet staining present in the destaining solution was measured at 590 nm. Wells filled with 200 μL of pure destaining solution were used as blank reference. All wells from the same biological replicate at a given time point were averaged to provide a single point and used for statistical analysis. Cell counting in Neubauer chambers ensured an initial cell concentration varying by less than 5%, and OD600nm measurements in the supernatant at all timepoints of incubation showed similar OD values over all biological replicates, confirming that growth rates were not driving the observed 2-fold difference in attached biomass after 30 h. Representative images of attachment were obtained after 24 h following the above protocol until the first drying stage. Two wells were imaged in their centre using phase contrast (Nikon Ti-E inverted microscope, 40× objective lens).

## Protein modelling

Protein sequences were entered into the AlphaFoldServer[94]. Pairwise structure alignment was conducted using the RCSB PDB web server with the TM-align method[95,96].

## Website design

Access to taxonomic, cultivation, and genomic information about the HiBC strains is available at https://www.hibc.rwth-aachen.de. This website was created using the Shiny package from R, and the code is available at: https://git.rwth-aachen.de/clavellab/hibc. The 16S rRNA gene sequence and genome for each strain can be downloaded directly from this site, both individually for strains of interest via the research data management platform Coscine, or for the entire collection via the digital repository Zenodo. Further strain-specific metadata provided includes: isolation conditions, source, and risk group.

## Description of novel taxa

The description of novel taxa was based on the analysis provided by Protologger v1.3[82], and manually curated into the protologues below. For each isolate, taxonomy was assigned using the following thresholds: <98.7% 16S rRNA gene sequence identity (as an indication for as-yet undescribed species), <94.5% (undescribed genus), and <86.5% (undescribed family)[97]. ANI values <95% to separate species[91]; POCP values <50% for distinct genera[98]. Phylogenomic trees were also considered to make decisions on genus- and family-level delineation[99]. All novel taxa have been registered with the SeqCode and will be registered with the ICNP.

## Description of *Alistipes intestinihominis* sp. nov.

*Alistipes intestinihominis* (in.tes.ti.ni.ho'mi.nis. L. neut. n. *intestinum*, the intestine; L. masc. n. *homo*, a human being; N.L. gen. n. *intestinihominis*, of the human intestine).

The genome size is 3.85 Mbp, G+C percentage is 58.29%, with 99.76% completeness and 0.96% contamination. Strain CLA-KB-H122 was determined to be a new species based on 16S rRNA gene analysis, with the closest validly named match being *Alistipes timonensis*

(98.17%). Separation from existing *Alistipes* species was confirmed by ANI comparison, which gave a value of 91.78% to *A. timonensis*. GTDB-Tk classified strain CLA-KB-H122 as 'Alistipes senegalensis'. However, the latter name is as yet not valid. Functional analysis showed the strain has 83 transporters, 17 secretion genes, and predicted utilisation of cellulose and starch along with production of L-glutamate and folate. In total, 395 CAZymes were identified, with 59 different glycoside hydrolase families and 19 glycoside transferase families represented. Major (≥5%) cellular fatty acids after 72 h of growth in DSMZ medium 1611 included 15:0 ISO FAME (19.4%), 15:0 FAME (18.9%), 16:0 ISO FAME (10.6%), 17:0 ISO 3OH FAME (8.8%), 17:0 FAME (7.2%), and 17:0 3OH FAME (6.9%). The type strain, CLA-KB-H122[T] (phylum *Bacteroidota*, family *Rikenellaceae*) (=DSM 118481) (StrainInfo: https://doi.org/10.60712/SI-ID414389.1, genome: GCA_040095975.1), was isolated from human faeces.

## Description of *Bifidobacterium hominis* sp. nov.

*Bifidobacterium hominis* (ho'mi.nis. L. gen. n. *hominis*, of a human being, pertaining to the human gut habitat, from where the type strain was isolated).

The genome size is 2.03 Mbp, G+C percentage is 55.98%, with 99.77% completeness and 0.45% contamination. The closest relative to strain CLA-AA-H311 was *Bifidobacterium pseudocatenulatum* (99.07%) based on 16S rRNA gene analysis. However, ANI comparison identified strain CLA-AA-H311 as a novel species within the genus *Bifidobacterium*, with an ANI value of 93.18% against *B. pseudocatenulatum*. GTDB-Tk classification as 'Bifidobacterium sp002742445' confirmed the proposition of a novel species. Placement within the genus *Bifidobacterium* was confirmed by the presence of fructose-6-phosphate phosphoketolase (KO1621)[100]. Functional analysis showed the strain has 90 transporters, 20 secretion genes, and predicted utilisation of starch and production of propionate, acetate, and folate. In total, 137 CAZymes were identified, with 28 different glycoside hydrolase families and 11 glycoside transferase families represented. Major (≥5%) cellular fatty acids after 24 h of growth in DSMZ medium 1203a included 16:0 FAME (28.8%), 18:0 FAME (15.3%), 18:1 CIS 9 DMA (15.1%), 18:1 CIS 9 FAME (14.3%), and 14:0 FAME (7.8%). The type strain, CLA-AA-H311[T] (phylum *Actinomycetota*, family *Bifidobacteriaceae*) (=DSM 118068, =LMG 33596) (StrainInfo: https://doi.org/10.60712/SI-ID414317.1, genome: GCA_040095915.1), was isolated from human faeces.

## Description of *Blautia aquisgranensis* sp. nov.

*Blautia aquisgranensis* (a.quis.gra.nen'sis. L. fem. adj. *aquisgranensis*, named after the German city of Aachen (Latin name *Aquisgranum*) where it was isolated).

The genome size is 3.63 Mbp, G+C percentage is 43.76%, with 99.37% completeness and 0.32% contamination. It includes two plasmids (37,495 bp; 1036 bp). The closest relative to strain CLA-JM-H16 was *Blautia intestinalis* (96.11%) based on 16S rRNA gene analysis. Placement of the strain within *Blautia* was confirmed based on POCP comparison, as values above 50% to multiple *Blautia* species were obtained. However, compared to the type species of the genus, *Blautia coccoides*, gave a value of 42.36%. This inconsistency was also highlighted by GTDB-Tk, which classified strain CLA-JM-H16 as 'Blautia_A sp900764225', suggesting *Blautia* may require splitting into multiple genera in future. As the separation of *Blautia* would require detailed analysis, which is outside the scope of this manuscript, we propose strain CLA-JM-H16 as a novel species within *Blautia*. All three novel species of *Blautia* described within this manuscript were confirmed to represent distinct species based on ANI comparison (*B. aquisgranensis* Vs. *B. caccae*, 81.74%; *B. aquisgranensis* Vs. *B. intestinihominis*, 76.95%; *B. caccae* Vs. *B. intestinihominis*, 78.57%). Functional analysis revealed 157 transporters, 17 secretion genes, and predicted utilisation of arbutin, salicin, sucrose, starch, and production of acetate, propionate, folate, L-glutamate, riboflavin, and cobalamin. In total, 205 CAZymes

were identified, with 40 different glycoside hydrolase families and 12 glycoside transferase families represented. The type strain, CLA-JM-H16$^T$ (phylum *Bacillota*, family *Lachnospiraceae*) (=DSM 114586, =LMG 33033) (StrainInfo: https://doi.org/10.60712/SI-ID414368.1, genome: GCA_040096615.1), was isolated from human faeces.

**Description of *Blautia caccae* sp. nov.** *Blautia caccae* (cac'cae. N.L. fem. n. *cacca*, human ordure, faeces; from Gr. fem. n. *kakkê*, human ordure, faeces; N.L. gen. n. *caccae*, of faeces, referring to the source of isolate).

The genome size is 5.83 Mbp, G+C percentage is 46.73%, with 98.73% completeness and 0.63% contamination. The closest relative to strain CLA-SR-H028 was *Blautia hominis* (98.66%) based on 16S rRNA gene analysis. ANI comparison identified CLA-SR-H028 as a novel species within the genus *Blautia*, with all values being below the species threshold. GTDB-Tk classification as 'Blautia sp001304935' confirmed the proposition of a novel species within *Blautia*. Functional analysis showed the strain has 158 transporters, 18 secretion genes, and predicted utilisation of cellobiose, sucrose, starch and production of propionate, acetate, cobalamin, and folate. In total, 353 CAZymes were identified, with 53 different glycoside hydrolase families and 15 glycoside transferase families represented. Major (≥5%) cellular fatty acids after 24 h of growth in DSMZ medium 1203a included 16:0 FAME (20.2%), 14:0 FAME (19.8%), 16:0 DMA (17.5%), 18:1 CIS 11 DMA (6.8%), and 14:0 DMA (5.9%). The type strain, CLA-SR-H028$^T$ (phylum *Bacillota*, family *Lachnospiraceae*) (=DSM 118556, =LMG 33609) (Straininfo: https://doi.org/10.60712/SI-ID414428.1, genome: GCA_040095955.1), was isolated from human faeces.

**Description of *Blautia intestinihominis* sp. nov.** *Blautia intestinihominis* (in.tes.ti.ni.ho'mi.nis. L. neut. n. *intestinum*, intestine; L. masc. n. *homo*, a human being; N.L. gen. n. *intestinihominis*, of the human intestine).

The genome size is 4.1 Mbp, G+C percentage is 43.49%, with 98.73% completeness and 0.63% contamination. It includes a single plasmid of 22,629 bp. The isolate was assigned to the species *Blautia obeum* (98.98%) based on 16S rRNA gene analysis. However, ANI comparison to *B. obeum* clearly identified this isolate as being a separate species (84.16%). This was confirmed by GTDB-Tk classification as 'Blautia_A sp000436615', recommending the creation of a novel species. Functional analysis showed the strain has 155 transporters, 18 secretion genes, and predicted utilisation of sucrose and starch, along with production of L-glutamate, folate, propionate, and cobalamin. In total, 177 CAZymes were identified. The type strain, CLA-AA-H95$^T$ (phylum *Bacillota*, family *Lachnospiraceae*) (=DSM 111354, =LMG 33582) (StrainInfo: https://doi.org/10.60712/SI-ID414326.1, genome: GCA_040096655.1), was isolated from human faeces.

**Description of *Coprococcus intestinihominis* sp. nov.** *Coprococcus intestinihominis* (in.tes.ti.ni.ho'mi.nis. L. neut. n. *intestinum*, intestine; L. masc. n. *homo*, a human being; N.L. gen. n. *intestinihominis*, of the human intestine).

The genome size is 3.6 Mbp, G+C percentage is 43.29%, with 98.43% completeness and 2.52% contamination. A single plasmid of 20,255 bp was detected. The closest relative to strain CLA-AA-H190 was *Coprococcus catus* (96.76%) based on 16S rRNA gene analysis. ANI comparison identified CLA-AA-H190 as a novel species within the genus *Coprococcus*, with an ANI value of 90.25% against the closest relative *C. catus*. GTDB-Tk classification as 'Coprococcus_A catus_A' confirmed the proposition of a novel species, but also suggests that the separation of *Coprococcus* into multiple genera could occur in future. Functional analysis showed the strain has 119 transporters, 18 secretion genes, and predicted utilisation of starch and production of propionate, butyrate, acetate, cobalamin, and folate. In total, 122 CAZymes were identified, with 19 different glycoside hydrolase families and 13 glycoside transferase families

represented. The type strain, CLA-AA-H190$^T$ (phylum *Bacillota*, family *Lachnospiraceae*) (=DSM 114688, =LMG 33015) (StrainInfo: https://doi.org/10.60712/SI-ID414125.1, genome: GCA_040096555.1), was isolated from human faeces.

**Description of *Enterobacter intestinihominis* sp. nov.** *Enterobacter intestinihominis* (in.tes.ti.ni.ho'mi.nis. L. neut. n. *intestinum*, the intestine; L. masc. n. *homo*, a human being; N.L. gen. n. *intestinihominis*, of the human intestine).

The genome size is 4.86 Mbp, G+C percentage is 54.88%, with 99.89% completeness and 0.12% contamination. It contains two plasmids (4416 bp; 2494 bp). Strain CLA-AC-H004 was determined to be a strain of *Enterobacter quasihormaechei* (99.80%) based on 16S rRNA gene analysis. Separation from existing *Enterobacter* species was confirmed by ANI comparison, which gave a value of 93.69% to *E. quasihormaechei*. GTDB-Tk classification of strain CLA-AC-H004 as 'Enterobacter hormaechei_A' supports the proposal of a novel species. An ANI value of 99.01% was obtained when compared to *Enterobacter hormaechei* subsp. *hoffmannii*. However, given the separation from *E. hormaechei* we propose these strains represent a separate species and not only a subspecies. ANI comparison also highlighted similarity with *Pedobacter himalayensis* (95.89%), which has been classified as 'Enterobacter hormaechei_B' within GTDB. However, this suggests reclassification of *Pedobacter* may be required in future. Functional analysis showed the strain has 497 transporters, 98 secretion genes, and predicted utilisation of arbutin, salicin, cellobiose, sucrose, and starch, along with production of L-glutamate, biotin, riboflavin, acetate, propionate, and folate. In total, 316 CAZymes were identified, with 37 different glycoside hydrolase families and 19 glycoside transferase families represented. Major (≥5%) cellular fatty acids after 24 h of growth in DSMZ medium 1203a included 16:0 FAME (38.3%), 17:0 CYCLO CIS 9,10 FAME (18.6%), 18:1 CIS 11 FAME (9.7%), 14:0 FAME (9.0%), 19:0 CYCLO CIS 11,12 FAME (9.0%), and 14:0 3OH (8.8%). The type strain, CLA-AC-H004$^T$ (phylum *Pseudomonadota*, family *Enterobacteriaceae*) (=DSM 118557, =LMG 33610) (StrainInfo: https://doi.org/10.60712/SI-ID414328.1, genome: GCA_040096145.1), was isolated from human faeces.

**Description of *Enterocloster hominis* sp. nov.** *Enterocloster hominis* (ho'mi.nis. L. gen. n. *hominis*, of a human being).

The genome size is 6.52 Mbp, G+C percentage is 50.14%, with 99.16% completeness and 2.53% contamination. It includes a single plasmid of 7635 bp. The closest relative to strain CLA-SR-H021 was *Enterocloster aldenensis* (98.44%) based on 16S rRNA gene analysis. ANI comparison identified CLA-SR-H021 as a novel species within the genus *Enterocloster*, with all values being below the species threshold. GTDB-Tk classified CLA-SR-H021 as 'Enterocloster pacaense', a name derived from the proposed species 'Lachnoclostridium pacaense'. However, the fact that these two names are not valid supports the proposition of a novel species within *Enterocloster*. Functional analysis revealed 118 transporters, 14 secretion genes, and predicted utilisation of cellobiose, starch and production of propionate, acetate, and folate. In total, 307 CAZymes were identified, with 39 different glycoside hydrolase families and 14 glycoside transferase families represented. Major (≥5%) cellular fatty acids after 24 h of growth in DSMZ medium 1203a included 16:0 FAME (30.3%), 16:1 CIS 9 DMA (13.7%), 16:1 CIS 9 FAME (10.7%), 14:0 FAME (10.4%), 16:0 DMA (9.6%), 18:1 CIS 11 DMA (5.6%), and 18:1 CIS 11 FAME (5.0%). The type strain, CLA-SR-H021$^T$ (phylum *Bacillota*, family *Lachnospiraceae*) (=DSM 118482, =LMG 33606) (StrainInfo: https://doi.org/10.60712/SI-ID414422.1, genome: GCA_040096035.1), was isolated from human faeces.

**Description of *Faecalibacterium intestinale* sp. nov.** *Faecalibacterium intestinale* (in.tes.ti.na'le. N.L. neut. adj. *intestinale*, pertaining to the intestine, from where the type strain was isolated).

The genome size is 2.97 Mbp, G+C percentage is 56.43%, with 100.0% completeness and 0.0% contamination. The isolate was determined to be related to *F. prausnitzii* (98.08%) based on 16S rRNA gene analysis. ANI comparison to *F. prausnitzii* was just below species-level assignment (94.46%), and GTDB-Tk classification as 'Faecalibacterium prausnitzii_J' recommended the creation of a novel species. Functional analysis showed the strain has 135 transporters, 18 secretion genes, and predicted utilisation of starch and production of L-glutamate, riboflavin, and cobalamin. In total, 159 CAZymes were identified, with 27 different glycoside hydrolase families and 12 glycoside transferase families represented. Production of butyrate (4.74 ± 0.30 mM) was confirmed for strain CLA-AA-H281 when grown in YCFA broth (DSMZ Medium No. 1611) in Hungate tubes for 48 h at 37 °C under anaerobic conditions (6% $CO_2$ and 4.7% $H_2$ in $N_2$). Major (≥5%) cellular fatty acids after 48 h of growth in DSMZ medium 215 included 16:0 FAME (33.2%), 18:1 CIS 11 DMA (11.7%), 18:1 CIS 11 FAME (11.4%), 14:0 FAME (9.3%), 16:0 DMA (8.4%), and 16:1 CIS 9 FAME (6.7%). The type strain, CLA-AA-H281[T] (phylum *Bacillota*, family *Oscillospiraceae*) (=DSM 116193, =LMG 33027) (StrainInfo: https://doi.org/10.60712/SI-ID414306.1, genome: GCA_040096575.1), was isolated from human faeces.

## Description of *Faecalibacterium tardum* sp. nov.

*Faecalibacterium tardum* (tar′dum. L. neut. adj. *tardum*, pertaining to its slow growth).

The genome size is 3.04 Mbp, G+C percentage is 56.28%, with 99.32% completeness and 0.0% contamination. It contains one plasmid of 14,735 bp in size, which encodes vancomycin resistance via the *vanY* gene. The isolate was determined to be closely related to *Faecalibacterium prausnitzii* (98.70%) based on 16S rRNA gene analysis. ANI comparison to *F. prausnitzii* was on the border of species-level assignment (95.05%), however, GTDB-Tk classification as 'Faecalibacterium prausnitzii_A' recommended the creation of a novel species. Strain CLA-AA-H175 was confirmed to represent a distinct species to *Faecalibacterium intestinale* (CLA-AA-H281, =DSM 116193) (ANI: 94.39%), also described in this paper. Functional analysis showed the strain has 116 transporters, 18 secretion genes, and predicted utilisation of starch and production of L-glutamate. In total, 177 CAZymes were identified, with 28 different glycoside hydrolase families and 12 glycoside transferase families represented. Production of butyrate (0.81 ± 0.16 mM) was confirmed for strain CLA-AA-H175 when grown in YCFA broth (DSMZ Medium No. 1611) in Hungate tubes for 48 h at 37 °C under anaerobic conditions (6% $CO_2$ and 4.7% $H_2$ in $N_2$). The type strain, CLA-AA-H175[T] (phylum *Bacillota*, family *Oscillospiraceae*) (=DSM 116192) (StrainInfo: https://doi.org/10.60712/SI-ID414281.1, genome: GCA_040096515.1), was isolated from human faeces.

## Description of *Faecousia* gen. nov.

Faecousia (Faec.ou′si.a. L. fem. n. *faex*, dregs; N.L. fem. n. *ousia*, an essence; N.L. fem. n. *Faecousia*, a microbe associated with faeces).

Based on 16S rRNA gene sequence identity, the closest relatives are members of the genus *Vescimonas* (*Vescimonas fastidiosa*, 93.48%). POCP analysis against *Vescimonas coprocola* (44.75%), the type species of this genus, and *V. fastidiosa* (41.06%) confirmed that strain CLA-AA-H192 represents a distinct genus from *Vescimonas*. GTDB-Tk supported the creation of a novel genus, placing strain CLA-AA-H192 within the proposed genus of "*Candidatus Faecousia*". The type species of this genus is *Faecousia intestinalis*.

## Description of *Faecousia intestinalis* sp. nov.

*Faecousia intestinalis* (in.tes.ti.na′lis. N.L. fem. adj. *intestinalis*, pertaining to the intestines, from where the type strain was isolated).

The genome size is 2.98 Mbp, G+C percentage is 58.44%, with 93.29% completeness and 0.0% contamination. It contains two plasmids (23,094 bp; 3448 bp). Functional analysis showed the strain has 144 transporters, 26 secretion genes, and predicted utilisation of

starch, and production of acetate, propionate, L-glutamate, and folate. In total, 116 CAZymes were identified, with 19 different glycoside hydrolase families and 11 glycoside transferase families represented. The type strain, CLA-AA-H192[T] (phylum *Bacillota*, family *Oscillospiraceae*) (StrainInfo: https://doi.org/10.60712/SI-ID414286.1, genome: GCA_040096185.1), was isolated from human faeces.

## Description of *Flavonifractor hominis* sp. nov.

*Flavonifractor hominis* (ho′mi.nis. L. gen. n. *hominis*, of a human being).

The genome size is 2.94 Mbp, G+C percentage is 58.64%, with 99.33% completeness and 0.0% contamination. Strain CLA-AP-H34 was determined to be a new species based on 16S rRNA gene sequence analysis, with the closest validly named match being *Flavonifractor plautii* (97.21%). Separation from existing *Flavonifractor* species was confirmed by ANI comparison, which gave a value of 82.25% to *F. plautii*. GTDB-Tk classification of strain CLA-AP-H34 as an unknown species within *Flavonifractor* supports the proposal of a novel species. Functional analysis showed the strain has 112 transporters, 14 secretion genes, and predicted utilisation of starch and production of L-glutamate, riboflavin, butyrate, and folate. In total, 140 CAZymes were identified, with 19 different glycoside hydrolase families and 15 glycoside transferase families represented. Major (≥5%) cellular fatty acids after 72 h of growth in DSMZ medium 1611 included 16:0 DMA (25.2%), 15:0 FAME (15.7%), 15:0 DMA (11.4%), 14:0 FAME (8.9%), 15:0 ISO FAME (8.4%), 17:0 DMA (7.5%), and 16:0 FAME (6.4%). The type strain, CLA-AP-H34[T] (phylum *Bacillota*, family *Oscillospiraceae*) (=DSM 118484, =LMG 33602) (StrainInfo: https://doi.org/10.60712/SI-ID414347.1, genome: GCA_040095835.1), was isolated from human faeces.

## Description of *Hominiventricola aquisgranensis* sp. nov.

*Hominiventricola aquisgranensis* (a.quis.gra.nen′sis. L. masc. adj. *aquisgranensis*, named after the German city of Aachen, where it was isolated).

The genome size is 3.17 Mbp, G+C percentage is 45.01%, with 99.37% completeness and 0.27% contamination, including a single plasmid of 7983 bp. The closest relative to strain CLA-AA-H78B was *Hominiventricola filiformis* (96.24%) based on 16S rRNA gene analysis. POCP analysis confirmed genus assignment to *Hominiventricola*, with a POCP value of 69.06% to the type strain of the only current species within this genus, *H. filiformis*. GTDB-Tk classified strain CLA-AA-H78B as 'Choladocola sp003480725' within "*Candidatus* Choladocola". The latter is a heterosynonym of *Hominiventricola*, a validly published name. ANI comparison confirmed that strain CLA-AA-H78B represents a novel species, as the ANI value to *H. filiformis* was 82.98%. Functional analysis showed the strain has 134 transporters, 31 secretion genes, and predicted utilisation of cellobiose, starch, arbutin, salicin, and production of L-glutamate, folate, acetate, propionate, and cobalamin. In total, 134 CAZymes were identified. The type strain, CLA-AA-H78B[T] (phylum *Bacillota*, family *Lachnospiraceae*) (=DSM 111355, =LMG 33583) (StrainInfo: https://doi.org/10.60712/SI-ID414323.1, genome: GCA_040096225.1), was isolated from human faeces.

## Description of *Lachnospira intestinalis* sp. nov.

*Lachnospira intestinalis* (in.tes.ti.na′lis. N.L. fem. adj. *intestinalis*, pertaining to the intestines, from where the type strain was isolated).

The genome size is 3.1 Mbp, G+C percentage is 41.75%, with 99.33% completeness and 0.0% contamination. Strain CLA-AA-H89B was determined to represent a separate species from its closest relative, *Lachnospira pectinoschiza* (97.48%), based on 16S rRNA gene analysis. This was confirmed based on ANI comparison to all close relatives, which were below the species threshold. GTDB-Tk classification of strain CLA-AA-H89B as 'Lachnospira sp000437735' confirmed that this isolate represents a novel species. Functional analysis showed the strain has 112 transporters, 29 secretion genes, and predicted utilisation of starch and cellulose, along with production of L-

glutamate, folate, acetate, propionate, and riboflavin. Motility was predicted based on the presence of the following genes: *FlhA, FlhB, FlgB, FlgC, FlgD, FlgE, FlgF, FlgJ, FlgK, FlgL, FliC, FliD, FliE, FliF, FliG, FliK, FliM, FliN, MotA, MotB*. In total, 162 CAZymes were identified, with 21 different glycoside hydrolase families and 11 glycoside transferase families represented. Major (≥5%) cellular fatty acids after 48 h of growth in DSMZ medium 1611 included 16:0 FAME (32.4%), 18:1 CIS 11 DMA (20.6%), 14:0 FAME (7.8%), 16:0 DMA (6.3%), 18:1 CIS 11 aldehyde (6.2%), and 16:1 CIS 9 DMA (5.5%). The type strain, CLA-AA-H89B[T] (phylum *Bacillota*, family *Lachnospiraceae*) (=DSM 118070) (StrainInfo: https://doi.org/10.60712/SI-ID414325.1, genome: GCA_040095895.1), was isolated from human faeces.

**Description of *Lachnospira hominis* sp. nov.** *Lachnospira hominis* (ho'mi.nis. L. gen. masc. n. *hominis*, of a human being).

The genome size is 3.15 Mbp, G+C percentage is 37.05%, with 98.66% completeness and 0.67% contamination. Strain CLA-JM-H10 was determined to represent a separate species from its closest relative, *Lachnospira rogosae* (96.37%), based on 16S rRNA gene analysis. This was confirmed by an ANI of 79.9% between *L. rogosa* (CLA-AA-H255) and strain CLA-JM-H10. GTDB-Tk classification of CLA-JM-H10 as 'Lachnospira sp900316325' confirmed that it represents a novel species. Functional analysis showed the strain has 109 transporters, 27 secretion genes, and predicted utilisation of starch, and production of L-glutamate, folate, and cobalamin. Motility was predicted based on detection of the following genes: *FlhA, FlhB, FlgB, FlgC, FlgD, FlgE, FlgF, FlgJ, FlgK, FlgL, FliC, FliD, FliE, FliF, FliG, FliK, FliM, FliN, MotA, MotB*. This is consistent with the type species of this genus being motile. In total, 165 CAZymes were identified, with 22 different glycoside hydrolase families and 12 glycoside transferase families represented. The type strain, CLA-JM-H10[T] (phylum *Bacillota*, family *Lachnospiraceae*) (=DSM 114599, =LMG 33585) (StrainInfo: https://doi.org/10.60712/SI-ID414366.1, genome: GCA_040096395.1), was isolated from human faeces.

**Description of *Lachnospira rogosae* sp. nov.** *Lachnospira rogosae* (ro.go'sae. N.L. gen. masc. n. *rogosae*, of Rogosa).

Strain CLA-AA-H255 was determined to be similar to *Lactobacillus rogosae* (99.87%) based on 16S rRNA gene analysis. However, the lack of a genome for the type strain of the latter species, along with the lack of the type strain at any established culture collection, prevented further comparison (Tindall, 2014). GTDB-Tk classification of CLA-AA-H255 as 'Lachnospira rogosae_A' suggested that it represents a species within a genus distantly related to *Lactobacillus*. This was reconfirmed by the same placement of a second strain, CLA-AA-H191, which was also assigned to the same placeholder by GTDB-Tk (ANI of 98.86% between our two isolates). Based on these results, we propose *L. rogosae* was previously misassigned as a member of the genus *Lactobacillus*. To provide a type strain and correct its placement, we propose the creation of the species *Lachnospira rogosae*. Functional analysis showed that strain CLA-AA-H255 has 100 transporters, 25 secretion genes, and predicted utilisation of starch, and production of L-glutamate, folate, and riboflavin. The prediction of motility based on genomic analysis is consistent with the observed phenotype of motility in the original type strain of *L. ragosae* as stated by Holdeman and Moore[101]. In total, 142 CAZymes were identified, including 20 different glycoside hydrolase families and 12 glycoside transferase families. Ecological analysis of 1000 human gut 16S rRNA gene amplicon samples identified this strain in 1.20% of samples with a relative abundance of 0.43 ± 0.78%. The type strain, CLA-AA-H255[T] (phylum *Bacillota*, family *Lachnospiraceae*) (=DSM 118602, =LMG 33594) (StrainInfo: https://doi.org/10.60712/SI-ID414300.1, genome: GCA_040096455.1), was isolated from human faeces.

**Description of *Laedolimicola intestinihominis* sp. nov.** *Laedolimicola intestinihominis* (in.tes.ti.ni.ho'mi.nis. L. neut. n. *intestinum*, intestine;

L. masc. n. *homo*, a human being; N.L. gen. n. *intestinihominis*, of the human intestine).

The genome size is 3.45 Mbp, G+C percentage is 49.65%, with 99.37% completeness and 0.16% contamination. It includes a single plasmid of 5131 bp. Strain CLA-AA-H132 was determined to represent a separate species from its closest relative, *Laedolimicola ammoniilytica* (98.38%), based on 16S rRNA gene analysis. POCP analysis confirmed that strain CLA-AA-H132 belongs to the recently named genus, *Laedolimicola*, with a POCP value of 75.19% to the type strain of the only current species within this genus, *L. ammoniilytica*. GTDB-Tk placement as 'Merdisoma sp900553635' suggests that *Laedolimicola* and *Candidatus* Merdisoma are homonyms and require future reclassification. ANI comparison confirmed that CLA-AA-H132 represents a novel species, as the ANI value to *L. ammoniilytica* was only 90.2%. Functional analysis showed the strain has 142 transporters, 21 secretion genes, and predicted utilisation of cellobiose, sucrose, starch, and production of acetate, propionate, L-glutamate, cobalamin, and folate. In total, 153 CAZymes were identified, with 21 different glycoside hydrolase families and 16 glycoside transferase families represented. Major (≥5%) cellular fatty acids after 24 h of growth in DSMZ medium 339 included 16:0 FAME (24.8%), 18:1 CIS 11 DMA (12.8%), 14:0 FAME (10.0%), 16:0 DMA (8.2%), and 16:0 CIS 9 DMA (8.2%). The type strain, CLA-AA-H132[T] (phylum *Bacillota*, family *Lachnospiraceae*) (=DSM 117481, =LMG 33588) (StrainInfo: https://doi.org/10.60712/SI-ID414271.1, genome: GCA_040096155.1), was isolated from human faeces.

**Description of *Maccoyibacter* gen. nov.** *Maccoyibacter* (Mac.co.y.i.-bac'ter N.L. fem. gen. n. *Maccoyae*, referring to the immunologist Kathy McCoy; N.L. masc. n. *bacter*, a rod; N.L. masc. n. *Maccoyibacter*, a rod-shaped bacterium named after the immunologist Kathy McCoy, for her many scientific contributions in the field of microbe-host interactions).

Based on 16S rRNA gene analysis, the closest species with a valid name are *Roseburia hominis* (94.9%) and *Eubacterium oxidoreducens* (94.23%). POCP values against all close relatives were below the genus delineation threshold of 50%, including many *Roseburia* spp. (*R. hominis, R. faecis, R. porci, R. intestinalis*), apart from *R. inulinivorans* (50.83%). GTDB-Tk placement as "UBA11774 sp003507655" confirmed that strain CLA-AA-H185 represents a novel genus within the family *Lachnospiraceae*. The type species is *Maccoyibacter intestinihominis*.

**Description of *Maccoyibacter intestinihominis* sp. nov.** *Maccoyibacter intestinihominis* (in.tes.ti.ni.ho'mi.nis. L. neut. n. *intestinum*, intestine; L. masc. n. *homo*, a human being; N.L. gen. n. *intestinihominis*, of the human intestine).

The genome size is 2.85 Mbp, G+C percentage is 41.17%, with 97.41% completeness and 0.45% contamination. It contains a single plasmid (2341 bp). Functional analysis showed the strain has 99 transporters, 35 secretion genes, and predicted utilisation of starch, and production of acetate, propionate, L-glutamate, cobalamin, folate, and riboflavin. In total, 110 CAZymes were identified, with 17 different glycoside hydrolase families and 13 glycoside transferase families represented. The type strain, CLA-AA-H185[T] (phylum *Bacillota*, family *Lachnospiraceae*) (=DSM 118601) (StrainInfo: https://doi.org/10.60712/SI-ID414284.1, genome: GCA_040096355.1), was isolated from human faeces.

**Description of *Megasphaera intestinihominis* sp. nov.** *Megasphaera intestinihominis* (in.tes.ti.ni.ho'mi.nis. L. gen. neut. n. *intestini*, of the intestine; L. gen. masc. n. *hominis*, of a human being; N.L. gen. masc. n. *intestinihominis*, of the human intestine).

The genome size is 2.38 Mbp, G+C percentage is 53.59%, with 100.00% completeness and 0.00% contamination. The closest relative to strain CLA-AA-H81 was *Megasphaera indica* (99.09%) based on 16S rRNA gene analysis. ANI comparison to all close relatives was below the species assignment threshold (highest to *Megasphaera elsdenii*,

90.78%). GTDB-Tk classification as 'Megasphaera sp000417505' supports the creation of a novel species. Functional analysis showed the strain has 133 transporters, 16 secretion genes, and predicted utilisation of starch, and production of butyrate, propionate, L-glutamate, folate, riboflavin, and cobalamin. Ecological analysis identified 152 matching (MASH distance <0.05) MAGs, of which 124 originate from the human gut, suggesting this species is most commonly observed within this environment. This is supported by ecological analysis using 16S rRNA gene amplicon datasets, which identified it within 19.0% of 1000 human gut samples, with a relative abundance of 2.14 ± 5.23%. Major (≥5%) cellular fatty acids after 24 h of growth in DSMZ medium 215 included 12:0 FAME (17.1%), 14:0 3OH (14.8%), 18:1 CIS 9 FAME (9.6%), 18:1 CIS 9 DMA (9.3%), 16:0 FAME (8.0%), and 16:1 CIS 7 FAME (5.0%). The type strain, CLA-AA-H81$^T$ (phylum *Bacillota*, family *Veillonellaceae*) (=DSM 118069, =LMG 33597) (StrainInfo: https://doi.org/10.60712/SI-ID414324.1, genome: GCA_040096415.1), was isolated from human faeces.

### Description of *Niallia hominis* sp. nov. *Niallia hominis* (ho'mi.nis. L. gen. n. *hominis*, of a human being).

The genome size is 4.9 Mbp, G+C percentage is 35.33%, with 92.24% completeness and 3.20% contamination. Strain CLA-SR-H024 was assigned to *Niallia circulans* (100%) based on 16S rRNA gene analysis. However, ANI suggested the isolate represents a novel species compared to all *Niallia* spp. (values were below 80%). GTDB-Tk classification of strain CLA-SR-H024 as 'Niallia sp001076885' confirmed that it represents a novel species. Functional analysis showed the strain has 244 transporters, 43 secretion genes, and predicted utilisation of arbutin, salicin, cellobiose, starch, and dextran, along with production of L-glutamate, folate, and cobalamin. In total, 294 CAZymes were identified, with 34 different glycoside hydrolase families and 14 glycoside transferase families represented. Major (≥5%) cellular fatty acids after 72 h of growth in DSMZ medium 1203a included 15:0 ANTEISO FAME (29.7%), 16:0 FAME (16.8%), 15:0 ISO FAME (14.9%), 16:0 ISO FAME (9.8%), 14:0 FAME (8.4%), 17:0 ANTEISO FAME (6.9%), and 14:0 ISO FAME (5.2%). The type strain, CLA-SR-H024$^T$ (phylum *Bacillota*, family *Bacillaceae*) (=DSM 118483) (StrainInfo: https://doi.org/10.60712/SI-ID414424.1, genome: GCA_040095995.1), was isolated from human faeces.

### Description of *Peptoniphilus hominis* sp. nov. *Peptoniphilus hominis* (ho'mi.nis. L. gen. n. *hominis*, of a human being).

The genome size is 1.99 Mbp, G+C percentage is 33.8%, with 99.3% completeness and 0.0% contamination. Strain CLA-SR-H025 was assigned to *Peptoniphilus gorbachii* (99.11%) based on 16S rRNA gene analysis. However, ANI suggested the isolate represents a novel species compared to *P. gorbachii*, giving a value of 88.92%. GTDB-Tk classification of strain CLA-SR-H025 as 'Peptoniphilus_A grossensis' confirmed that it represents a novel species, as the name 'Peptoniphilus grossensis' was proposed but never validated[102]. Functional analysis showed the strain has 12 transporters, 1 secretion gene, and no gut metabolic models were identified within the genome. In total, 59 CAZymes were identified, with 10 different glycoside hydrolase families and 8 glycoside transferase families represented. The type strain, CLA-SR-H025$^T$ (phylum *Bacillota*, family *Peptoniphilaceae*) (=DSM 118555, =LMG 33608) (Straininfo: https://doi.org/10.60712/SI-ID414425.1, genome: GCA_040096015.1), was isolated from human faeces.

### Description of *Pseudoflavonifractor intestinihominis* sp. nov. *Pseudoflavonifractor intestinihominis* (in.tes.ti.ni.ho'mi.nis. L. neut. n. *intestinum*, the intestine; L. masc. n. *homo*, a human being; N.L. gen. n. *intestinihominis*, of the human intestine).

The genome size is 3.69 Mbp, G+C percentage is 61.54%, with 99.33% completeness and 0.00% contamination. Strain CLA-AP-H29 was assigned to the species *Pseudoflavonifractor capillosus* (99.53%)

based on 16S rRNA gene sequence analysis. However, ANI suggested the isolate represents a novel species compared to *P. capillosus*, giving a value of 80.96%. GTDB-Tk classified strain CLA-AP-H29 as 'Pseudoflavonifractor sp944387275', which confirms that it represents a novel species. Functional analysis predicted that the strain has 121 transporters, 14 secretion genes, and predicted utilisation of starch, and production of L-glutamate, folate, and cobalamin. In total, 180 CAZymes were identified, with 25 different glycoside hydrolase families and 15 glycoside transferase families represented. Major (≥5%) cellular fatty acids after 48 h of growth in DSMZ medium 215 included 14:0 FAME (33.9%), 16:0 DMA (33.8%), 16:0 FAME (9.8%), and 16:0 ALDE (8.0%). The type strain, CLA-AP-H29$^T$ (phylum *Bacillota*, family *Oscillospiraceae*) (=DSM 118073, =LMG 33601) (Straininfo: https://doi.org/10.60712/SI-ID414342.1, genome: GCA_040096055.1), was isolated from human faeces.

### Description of *Robertmurraya yapensis* sp. nov. *Robertmurraya yapensis* (yap'ensis. N.L. fem. adj. *yapensis*, pertaining to Yap trench, which is the geographical position where the first isolate of this species was obtained).

The genome size is 4.74 Mbp, G+C percentage is 37.94%, with 98.85% completeness and 2.13% contamination. It contains a plasmid (2470 bp). The closest relative to strain CLA-AA-H227 was *Robertmurraya spiralis* (99.23%) based on 16S rRNA gene analysis. The highest POCP scores were for members of the genus *Robertmurraya (Robertmurraya kyonggiensis*, 84.9%; *R. spiralis*, 65.33%). The placement of strain CLA-AA-H227 within *Robertmurraya* was confirmed by GTDB-Tk assignment as 'Robertmurraya yapensis', a reclassification of the species 'Bacillus yapensis', although neither name has been validated. ANI comparison confirmed that CLA-AA-H227 represents a novel species, as values to close relatives were below the species-level threshold. Functional analysis showed the strain has 246 transporters, 53 secretion genes, and predicted utilisation of arbutin, salicin, cellobiose, starch, and production of butyrate, acetate, propionate, folate, riboflavin, and cobalamin. In total, 238 CAZymes were identified. The type strain, CLA-AA-H227$^T$ (phylum *Bacillota*, family *Bacillaceae*) (=DSM 113004, =LMG 33018) (Straininfo: https://doi.org/10.60712/SI-ID414150.1, genome: GCA_040096375.1), was isolated from human faeces.

### Description of *Ruminococcoides intestinale* sp. nov. *Ruminococcoides intestinale* (in.tes.ti.na'le. N.L. neut. adj. *intestinalis*, pertaining to the intestines, from where the type strain was isolated).

The genome size is 2.32 Mbp, G+C percentage is 40.88%, with 99.33% completeness and 1.01% contamination. The isolate was determined to be similar to *Ruminococcus bromii* (98.91%) and more distantly related to *Ruminococcoides bili* (96.76%) based on 16S rRNA gene analysis. While POCP comparison of strain CLA-JM-H38 to *R. bromii* was 59.79%, and 53.56% to *Ruminococcus bovis*, suggesting they belong to the same genus, all other comparisons to *Ruminococcus* species were below 50%, including to the type species, *Ruminococcus flavefaciens* (27.58%). POCP to *R. bili*, the type species of the genus *Ruminococcoides*, was 64.44%. GTDB-Tk classified strain CLA-JM-H38 as "Ruminococcus_E bromii_B", confirming it is not a member of the genus *Ruminococcus*. These results support the GTDB assignment that both *R. bovis* and *R. bromii* should be reclassified as members of the genus *Ruminococcoides*. Strain CLA-JM-H38 was confirmed to represent a novel species as all ANI comparisons to close relatives were below 95%, and it represents a distinct novel species from *Ruminococcoides intestinihominis* described in this work (78.33%). Functional analysis showed the strain has 81 transporters, 15 secretion genes, and predicted utilisation of starch, and production of L-glutamate. In total, 108 CAZymes were identified, with 15 different glycoside hydrolase families and 12 glycoside transferase families represented. Ecological analysis based on 16S rRNA gene amplicons identified this species in 55.20% of 1000 human gut samples with a relative abundance of

1.50 ± 2.49%, suggesting it is a prevalent and dominant bacterial species within the human gut. Major (≥5%) cellular fatty acids after 72 h of growth in DSMZ medium 1611 included 16:0 ISO FAME (49.5%) and 16:0 ISO DMA (23.8%). The type strain, CLA-JM-H38^T (phylum *Bacillota*, family *Oscillospiraceae*) (=DSM 118486, =LMG 33604) (StrainInfo: https://doi.org/10.60712/SI-ID414376.1, genome: GCA_040096305.1), was isolated from human faeces.

**Description of *Ruminococcoides intestinihominis* sp. nov.** *Ruminococcoides intestinihominis* (in.tes.ti.ni.ho'mi.nis. L. neut. n. *intestinum*, intestine; L. masc. n. *homo*, a human being; N.L. gen. n. *intestinihominis*, of the human intestine).

The genome size is 2.26 Mbp, G+C percentage is 34.26%, with 98.66% completeness and 0.00% contamination. It contains one plasmid (1825 bp). Based on 16S rRNA gene sequence similarity, the isolate was closely related to *Ruminococcus bovis* (99.3%) and more distant to *Ruminococcoides bili* (94.8%), the type species of this genus. ANI comparison confirmed the similarity of strain CLA-AA-H171 to *R. bovis* (95.4%); however, classification by GTDB-Tk as 'Ruminococcus_E sp934476515' supported the creation of a novel species. Recently, 'Ruminococcus_E' has been validly named as *Ruminococcoides*, with the type species *R. bili*[28]. POCP comparison between the isolate and *R. bili* provided a value of 51.3%, suggesting that strain CLA-AA-H171 represents a novel species within the genus *Ruminococcoides*. Functional analysis showed the strain has 75 transporters, 14 secretion genes, and predicted utilisation of starch and production of acetate. In total, 124 CAZymes were identified, with 13 different glycoside hydrolase families and 12 glycoside transferase families represented. Ecological analysis based on 16S rRNA gene amplicons identified this species in 10.40% of 1000 human gut samples with a relative abundance of 0.23 ± 0.68%. The type strain, CLA-AA-H171^T (phylum *Bacillota*, family *Oscillospiraceae*) (=DSM 114689, =LMG 33587) (StrainInfo: https://doi.org/10.60712/SI-ID414279.1, genome: GCA_040096285.1), was isolated from human faeces.

**Description of *Ruthenibacterium intestinale* sp. nov.** *Ruthenibacterium intestinale* (in.tes.ti.na'le. N.L. neut. adj. *intestinale*, pertaining to the intestines, from where the type strain was isolated).

The genome size is 3.1 Mbp, G+C percentage is 54.69%, with 98.3% completeness and 0.00% contamination. It contains a plasmid (5063 bp). The closest relative to strain CLA-JM-H11 is *Ruthenibacterium lactatiformans* (94.93%), the type species of this genus, based on 16S rRNA gene analysis. Placement of the isolate within the genus *Ruthenibacterium* was confirmed based on POCP comparison, with a value of 56.41% to the type species. Strain CLA-JM-H11 was confirmed to be distinct from *R. lactatiformans* based on ANI comparison (79.28%). GTDB-Tk classification confirmed this assignment as an unknown species within *Ruthenibacterium*. Functional analysis showed the strain has 143 transporters, 16 secretion genes, and predicted utilisation of starch, and production of acetate, propionate, folate, and cobalamin. In total, 155 CAZymes were identified, with 27 different glycoside hydrolase families and 13 glycoside transferase families represented. The type strain, CLA-JM-H11^T (phylum *Bacillota*, family *Oscillospiraceae*) (=DSM 114604, =LMG 33032) (StrainInfo: https://doi.org/10.60712/SI-ID414367.1, genome: GCA_040096265.1), was isolated from human faeces.

**Description of *Solibaculum intestinale* sp. nov.** *Solibaculum intestinale* (in.tes.ti.na'le. N.L. neut. adj. *intestinale*, pertaining to the intestines, from where the type strain was isolated).

The genome size is 2.81 Mbp, G+C percentage is 54.62%, with 97.99% completeness and 0.67% contamination. The isolate was determined to be a new species based on 16S rRNA gene analysis, with the closest validly named match being *Solibaculum mannosilyticum* (94.93%). Placement within *Solibaculum* was confirmed with POCP

values above 50% to *S. mannosilyticum* (55.7%), the type species, and only member of this genus. ANI comparison confirmed the isolate represents a novel species, as no ANI values were above 80%. Functional analysis showed the strain has 82 transporters, 16 secretion genes, and predicted utilisation of starch, and production of acetate, propionate, and L-glutamate. In total, 169 CAZymes were identified, with 20 different glycoside hydrolase families and 15 glycoside transferase families represented. The type strain, CLA-JM-H44^T (phylum *Bacillota*, family *Oscillospiraceae*) (=DSM 114601, =LMG 33034) (StrainInfo: https://doi.org/10.60712/SI-ID414377.1, genome: GCA_040096205.1), was isolated from human faeces.

**Description of *Ventrimonas* gen. nov.** *Ventrimonas* (Ven.tri.mo'nas. L. masc. n. *venter*, the belly; L. fem. n. *monas*, a monad; N.L. fem. n. *Ventrimonas*, a microbe associated with the belly (intestines/faeces)).

Based on 16S rRNA gene sequence similarity, the closest relatives are members of the genus *Hungatella* (*Hungatella effluvii*, 95.15%; *Hungatella hathewayi*, 94.95%). POCP analysis against *H. effluvii* (40.38%) and *H. hathewayi* (39.86%) indicates that strain CLA-AP-H27 represents a distinct genus within *Hungatella*. GTDB-Tk supported the creation of a novel genus, placing strain CLA-AP-H27 within the proposed genus of "*Candidatus* Ventrimonas" in the family *Lachnospiraceae*. The type species of this genus is *Ventrimonas faecis*.

**Description of *Ventrimonas faecis* sp. nov.** *Ventrimonas faecis* (fae'cis. L. gen. n. *faecis*, of dregs, pertaining to faeces, from where the type strain was isolated).

The genome size is 3.41 Mbp, G+C percentage is 49.27%, with 98.73% completeness and 0.63% contamination. Functional analysis showed the strain has 156 transporters, 21 secretion genes, and predicted utilisation of starch, and production of acetate, propionate, L-glutamate, cobalamin, and folate. In total, 139 CAZymes were identified, with 21 different glycoside hydrolase families and 12 glycoside transferase families represented. Major (≥10%) cellular fatty acids after 24 h of growth in DSMZ medium 1203a included 16:0 FAME (35.5%), 18:1 CIS 11 DMA (10.0%), 18:1 CIS 11 FAME (9.6%), 18:1 CIS 9 DMA (5.9%), and 16:0 DMA (5.5%). The type strain, CLA-AP-H27^T (phylum *Bacillota*, family *Lachnospiraceae*) (=DSM 118072, =LMG 33600) (StrainInfo: https://doi.org/10.60712/SI-ID414341.1, genome: GCA_040096075.1), was isolated from human faeces.

**Description of *Waltera hominis* sp. nov.** *Waltera hominis* (ho'mi.nis. L. gen. masc. n. *hominis*, of a human being, pertaining to the human gut habitat, from where the type strain was isolated).

The genome size is 3.88 Mbp, G+C percentage is 45.72%, with 99.52% completeness and 2.13% contamination. The isolate was assigned to the species *Waltera intestinalis* (100.0%) based on 16S rRNA gene analysis. However, ANI comparison identified strain CLA-AA-H183 as a novel species within the genus *Waltera*, with an ANI value of 91.97% against the type species *Waltera intestinalis*. GTDB-Tk currently lacks the inclusion of *Waltera*, causing misclassification as 'Acetatifactor sp003447295'; however, this confirmed the proposition of a novel species. Separation of *Waltera* from *Acetatifactor* was revalidated via both phylogenomic analysis (Supplementary Fig. 3), which shows both genera form separate monophyletic groups, and POCP analysis, which shows clear similarity within each genus (*Waltera*: 68.86%, *Acetatifactor*: 59.25%), and separation between the genera (39.94 ± 1.63%). Functional analysis showed the strain has 141 transporters, 36 secretion genes, and predicted utilisation of starch, cellulose, and production of butyrate, propionate, acetate, and folate. In total, 228 CAZymes were identified, with 38 different glycoside hydrolase families and 14 glycoside transferase families represented. The type strain, CLA-AA-H183^T (phylum *Bacillota*, family *Lachnospiraceae*) (=DSM 114684, =LMG 33586) (StrainInfo: https://doi.org/10.60712/SI-ID414283.1, genome: GCA_040096245.1), was isolated from human faeces.

**Reporting summary**

Further information on research design is available in the Nature Portfolio Reporting Summary linked to this article.

## Data availability

The genomes for all strains have been deposited at NCBI under BioProject PRJNA996881. Bulk download of HiBC resources is possible via Zenodo for the genomes (https://doi.org/10.5281/zenodo.12180083), plasmid sequences (https://doi.org/10.5281/zenodo.12187897), 16S rRNA gene sequences (https://doi.org/10.5281/zenodo.12180259), and the isolates metadata (https://doi.org/10.5281/zenodo.12180506). The PacBio genome for *P. vulgatus* CLA-AA-H253 (=DSM 118718) has been deposited in ENA under accession code PRJEB80480 and Zenodo (https://doi.org/10.5281/zenodo.14674027).

## Code availability

The codes for genome processing and creation of the HiBC website are available at https://github.com/ClavelLab/genome-assembly (https://doi.org/10.5281/zenodo.15172576) and https://git.rwth-aachen.de/clavellab/hibc (https://doi.org/10.5281/zenodo.15172609), respectively.

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

## Acknowledgements

Sequencing was performed with the support of: (i) the DFG-funded NGS Competence Center Tübingen (INST 37/1049-1) and the Institute for Medical Microbiology and Hygiene at the University Hospital (Tübingen, Germany), including help by the Quantitative Biology Center (QBiC) for raw data management and storage; (ii) Nassos Typas and Carlos Geert Pieter Voogdt (EMBL, Heidelberg), for long-read sequencing; (iii) the Genomics Facility, a core facility of the Interdisciplinary Center for Clinical Research (IZKF) Aachen within the Faculty of Medicine at RWTH Aachen University. We are also thankful to: (iv) Patrick Buchta from the Audio-Visual department at the University Hospital of RWTH Aachen for designing the HiBC logo; (v) Marzena Wyschkon and Meina Neumann-Schaal from the Leibniz Institute DSMZ for their help with the deposition of strains and CFA analysis, respectively; (vi) Catherine Gonzalez and Maurice Heizer from the Functional Microbiome Research Group (Institute of Medical Microbiology; University Hospital of RWTH Aachen) for data management with Coscine and for the curation of HiBC, respectively; (vii) Peter Vandamme, Claudine Vereecke, and Anneleen Wieme for handling the deposition of strains at the BCCM/LMG Bacteria Collection. T.C.A.H. received funding from the German Research Foundation (DFG) as part of SFB1382 (Rising Star programme) and from the RWTH Aachen START programme, project "LeakyGut". T.C. received funding from the DFG, project no. 513892404, no. 445552570, no. 395357507 – SFB1371, no. 403224013 – SFB1382 and no. 460129525 – NFDI4Microbiota, and the German Ministry for Research and Education (BMBF), project Mi-EOCRC (K.Z. 01KD2102D). J.O. received funding from the DFG, project no. 6270054 NFDI 28/1 "NFDI4Microbiota" and 6270048 NFDI 5/1 "NFDI4-Biodiversity", the BMBF, projects no. 8005512901 and 8005512001 of DZIF, and EU/Horizon IRA project MICROBE no. 101094353. M.G. received funding from the DFG, project no. 403224013 – SFB1382. The data used in this publication was managed using the research data management platform Coscine with storage space granted by the Research Data Storage (RDS) of the DFG and Ministry of Culture and Science of the State of North Rhine-Westphalia (DFG: INST222/1261-1 and MKW: 214-4.06.05.08 - 139057). L.M. and T.A. received funding from The Leverhulme Trust: project no. RF-2023-286\2.

## Author contributions

The project was conceptualised and managed by T.C.A.H. and T.C. Isolation and cultivation of bacteria were conducted by J.M.M., J.B., S.N., M.B., S.R., A.A., A.M.A., D.W., A.C., S.A.V.J., A.P., A.V., M.S., E.C.D., K.B., M.W., K.A.W., L.A., and A.R. M.B. performed analysis of *Bacteroidales* plasmids. Confirmation of plasmid pMMCAT presence was conducted by L.M. and T.A., with M.D.C. and M.G. evaluating its impact on adhesion. Cultivation of strains was supervised by T.C.A.H., D.B., T.S., and T.C., with samples for isolation from S.T., Th.Cr., and J.S. N.K. and N.T. performed genome sequencing. T.C.A.H., J.M.M., N.T., A.A., and N.K. conducted stool sample acquisition. Deposition of strains at the DSMZ was handled by S.N., A.L., I.S., J.W., T.R., M.P., B.A., L.C.R., and J.O. J.M.M., C.P., S.N., M.B., D.W., A.C., T.R., B.A., and T.C. curated isolates. The website and metadata curation were conducted by T.C.A.H., J.M.M., C.P., J.H., and L.C.B. Bioinformatic analysis was conducted by T.C.A.H., C.P., and M.A.S. C.P. deposited the sequencing data. T.C. coordinated the project.

J.O. and T.C. provided essential infrastructure and secured funding. The manuscript was written by T.C.A.H. and T.C., and reviewed by all authors.

## Funding

## Competing interests
The authors declare no competing interests.

## Additional information

[1]Functional Microbiome Research Group, Institute of Medical Microbiology, University Hospital of RWTH Aachen, Aachen, Germany. [2]Biophysics of Host-Microbe Interactions Research Group, Institute of Medical Microbiology, University Hospital of RWTH Aachen, Aachen, Germany. [3]Leibniz Institute DSMZ-German Collection of Microorganisms and Cell Cultures, Braunschweig, Germany. [4]Molecular Tumor Biology Research Group, Department of General, Visceral, Children and Transplantation Surgery, University Hospital of RWTH Aachen, Aachen, Germany. [5]German Centre for Infection Research (DZIF), Partner Site Hannover-Braunschweig, Braunschweig, Germany. [6]Department of Microbial Immune Regulation, Helmholtz Centre for Infection Research, Braunschweig, Germany. [7]Center for Microbiology and Environmental Systems Science, Department of Microbiology and Ecosystem Science, University of Vienna, Vienna, Austria. [8]Chair of Nutrition and Immunology, School of Life Sciences, Technical University of Munich, Freising-Weihenstephan, Germany. [9]Institute of Neuroanatomy, University Hospital of RWTH Aachen, Aachen, Germany. [10]School of Life Sciences, University of Nottingham, Nottingham, UK. [11]IT centre, RWTH Aachen University, Aachen, Germany. [12]Clinic for Child and Adolescent Psychiatry, Psychosomatic Medicine and Psychotherapy, LVR-University Hospital Essen, University of Duisburg-Essen, Essen, Germany. [13]Centre for Individualised Infection Medicine (CiiM), a joint venture between the Helmholtz-Centre for Infection Research (HZI) and the Hannover Medical School (MHH), Hannover, Germany. [14]Technical University Braunschweig, Braunschweig, Germany. [15]These authors contributed equally: Johannes M. Masson, Charlie Pauvert. ✉e-mail: tclavel@ukaachen.de

