## [Transparent Peer Review file · Nature Communications]

HiBC: a publicly available collection of bacterial strains isolated from the human gut

Corresponding Author: Professor Thomas Clavel

Version 0:

Reviewer comments:

Reviewer #1

(Remarks to the Author)

This is a much improved and clearer manuscript. The authors have addressed most of my comments. I would like them to address the following though:

Figure 1:

My comment:

Line 142- the more likely explanation is that marker genes in phylophlan are not robust for correct placement of fusobacterium genome. In addition, having a single genome from one phylum can also lead to incorrect phylogenetic placement. To resolve this the authors should try a different tool to generate phylogeny e.g. gtdb-tk

Authors response: There seems to be a misunderstanding here; Phylophlan was used to construct the tree, but our taxonomic assignments are manually curated from a mixture of methods, including GTDB-Tk. This is why we mentioned GTDB in the legend when referring to the 'Bacillota_A'. To avoid similar confusion with future readers, we have rephrased the legend as follows: "The potential need for splitting the phylum Bacillota is therefore independently supported by the Phylophlan tree, and GTDB."

I understand what the author's did, I am asking for validation of their approach. A different phylogenetic software needs to be used to confirm the placement of the single Fusobacterium species within the Bacillota and to confirm it is not mis-placed by Phylophlan. The obvious one to use, given the tree is comprised of different species is the GTDB-tk workflow identify-> align -> infer.

Figure 2:

Line 141 and Figure 2. Checking some of the papers used in the authors analysis, the number of strains 'requestable' is greater than that reported by the authors. For example, All HBC (Supplemental Table 1 of this paper has culture collection source, either public or the authors' lab) and BIO-ML strains (<https://www.broadinstitute.org/infectious-disease-and-microbiome/broad-institute-openbiome-microbiome-library>) are requestable . This needs to be corrected in the text and Figure 2.

(Remarks on code availability)

Reviewer #2

(Remarks to the Author)

(Remarks on code availability)

Reviewer #3

(Remarks to the Author)

I spent some time with the revised paper and I still feel that it is very unfocused. The response to reviewers document suggested that the authors had taken significant actions to focus and improve the paper but I don't see significant improvement in the paper. I've included some comments on few of my previous points below, but I still think the paper lacks focus and flow.

The abstract emphasizes a problem, which is that there are no bulk deposition systems for culture collections and then states that a bulk submission has been set up. But then the rest of the paper is not about the bulk submission system. Given this, the writing needs to change to introduce a different problem.

The response to reviewers states that a paper about a bulk strain submission system would be appropriate for Nature Protocols, but the problem introduced in the paper is that there is no bulk submission system and the following "solution" sentence in the abstract says they are introducing a bulk submission system.

I recommended before that the text on plasmids was completely disconnected from the problem/solution introduced in the abstract and introduction. The response to authors says that most information about plasmids was mostly removed from the paper, but in fact the revised paper mentions plasmids 105 times and has three sentences (>30%) on plasmids in the abstract, and dedicates Figure 3 entirely to plasmids.

Sorry but my review is basically the same as previously. The authors need to decide what the paper is about and then focus the entire paper on that subject.

(Remarks on code availability)

Reviewer #1 (Remarks to the Author):

This is a much improved and clearer manuscript. The authors have addressed most of my comments. I would like them to address the following though:

Figure 1:

My comment:

Line 142- the more likely explanation is that marker genes in phylophlan are not robust for correct placement of fusobacterium genome. In addition, having a single genome from one phylum can also lead to incorrect phylogenetic placement. To resolve this the authors should try a different tool to generate phylogeny e.g. gtdb-tk

Authors response: There seems to be a misunderstanding here; Phylophlan was used to construct the tree, but our taxonomic assignments are manually curated from a mixture of methods, including GTDB-Tk. This is why we mentioned GTDB in the legend when referring to the 'Bacillota_A'. To avoid similar confusion with future readers, we have rephrased the legend as follows: "The potential need for splitting the phylum Bacillota is therefore independently supported by the Phylophlan tree, and GTDB."

I understand what the author's did, I am asking for validation of their approach. A different phylogenetic software needs to be used to confirm the placement of the single Fusobacterium species within the Bacillota and to confirm it is not mis-placed by Phylophlan. The obvious one to use, given the tree is comprised of different species is the GTDB-tk workflow identify-> align -> infer.

Response: To validate the output of phylophlan we have used the GTDB-Tk workflow, as suggested by this reviewer. The additional tree (Supplementary Figure 4) shows no separation of Bacillota_A from Bacillota. We have included a brief text on these results in the Supplementary Results, detailing that the two trees contradict each other. Surprisingly, the GTDB-Tk inferred tree contradicts the suggested separation by the GTDB assignment of the isolates. A note specifying that the placement of these isolates depends on the method used for tree creation was added to the legend of figure 1 in the main text. We think unclear phylogenomic placement of a few genomes due to method is however not a major issue in the context of this broad collection of isolates and transparent information is provided in this revision.

Figure 2:

Line 141 and Figure 2. Checking some of the papers used in the authors analysis, the number of strains 'requestable' is greater than that reported by the authors. For example, All HBC (Supplemental Table 1 of this paper has culture collection source, either public or the authors' lab) and BIO-ML strains (<https://www.broadinstitute.org/infectious-disease-and-microbiome/broad-institute-openbiome-microbiome-library>) are requestable. This needs to be corrected in the text and Figure 2.

Response: We are thankful for the reviewer for pointing out this discrepancy. We have now checked the results, and modified both the Supplementary Table and Figure accordingly. While BIO-ML was always marked as all strains being requestable, the HBC strains are also identified as such. We also noticed the Supplementary Table stated "None responsive" for the hGMB collection, which whilst true, they do claim strains are requestable in their paper, hence we have edited this to state all 1,170 isolates are requestable. We have also edited the text in the paper to clarify that 76.1% of strains are requestable. Furthermore, we have refined the definition of requestable

within the paper to clarify that these claims were not validated: “ Strains were deemed requestable if it was claimed in the original publication, although these claims were not substantiated.”

Reviewer #2 (Remarks to the Author):

n/a

Reviewer #3 (Remarks to the Author):

I spent some time with the revised paper and I still feel that it is very unfocused. The response to reviewers document suggested that the authors had taken significant actions to focus and improve the paper but I don't see significant improvement in the paper. I've included some comments on few of my previous points below, but I still think the paper lacks focus and flow.

The abstract emphasizes a problem, which is that there are no bulk deposition systems for culture collections and then states that a bulk submission has been set up. But then the rest of the paper is not about the bulk submission system. Given this, the writing needs to change to introduce a different problem.

The response to reviewers states that a paper about a bulk strain submission system would be appropriate for Nature Protocols, but the problem introduced in the paper is that there is no bulk submission system and the following "solution" sentence in the abstract says they are introducing a bulk submission system.

I recommended before that the text on plasmids was completely disconnected from the problem/solution introduced in the abstract and introduction. The response to authors says that most information about plasmids was mostly removed from the paper, but in fact the revised paper mentions plasmids 105 times and has three sentences (>30%) on plasmids in the abstract, and dedicates Figure 3 entirely to plasmids.

Sorry but my review is basically the same as previously. The authors need to decide what the paper is about and then focus the entire paper on that subject.

Response: As discussed, we have removed the strain deposition system from the introduction and results of the paper. Within the discussion, we state that the system was initiated but provide only broad strokes of how it worked, which is instead detailed in the methods.

We have also modified the abstract and title to focus on the strain collection and removed any mention of the strain deposition system.